# Keep Everyone Happy: Online Fair Division of Numerous Items with Few Copies

**Arun Verma** [1]  **Indrajit Saha** [2]  **Makoto Yokoo** [3]  **Bryan Kian Hsiang Low** [4][1]

## Abstract

This paper considers a novel variant of the online fair division problem involving multiple agents in which a learner sequentially observes an indivisible item that must be irrevocably allocated to one of the agents to achieve a desired balance between fairness and efficiency. Existing algorithms assume a small number of items with a sufficiently large number of copies, which ensures a good utility estimation for all item-agent pairs from noisy observed utilities. However, this assumption may not hold in many real-life applications, e.g., an online platform with a large number of users (items) who use the platform's service providers (agents) only a few times (a few copies of items), making it difficult to accurately estimate utilities for all item-agent pairs. To address this limitation, we assume utility is an unknown function of item-agent features. We propose algorithms that model online fair division as a contextual bandit problem, achieving provable sub-linear regret. Our experimental results further validate the effectiveness of the proposed algorithms. The code is publicly available in this GitHub repository.

## 1. Introduction

Growing economic, environmental, and social pressures require us to be efficient with limited resources (Aleksandrov & Walsh, 2020). Therefore, the fair division (Steinhaus, 1948) of limited resources among multiple parties/agents is needed to efficiently balance their competing interests in many real-life applications, for example, Fisher markets (Codenotti & Varadarajan, 2007; Vazirani, 2007), housing allocations (Benabbou et al., 2019), rent division (Edward Su,

1999; Gal et al., 2016), and many more (Demko & Hill, 1988). The fair division problem has been extensively studied in algorithmic game theory (Eisenberg & Gale, 1959; Codenotti & Varadarajan, 2007; Vazirani, 2007; Caragiannis et al., 2019), but it focuses on the static setting where all information (items, agents, and their utilities) is known in advance. However, many real-life fair division problems are online (Aleksandrov & Walsh, 2020; Gao et al., 2021; Gkatzelis et al., 2021; Benadè et al., 2022; Liao et al., 2022; Yamada et al., 2024; Yang et al., 2024), referred to as ***online fair division***, in which indivisible items arrive sequentially and each item must be irrevocably allocated to an agent.

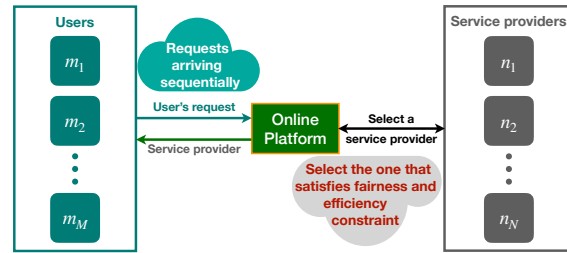

*Figure 1.* Example of an online fair division of numerous items with few copies: An online platform recommends service providers (agents) to users (items) who arrive sequentially over time. The platform must balance two inherently conflicting objectives: ensuring fair exposure and opportunity among service providers with competing interests (fairness), while simultaneously maximizing its own profit or overall system utility (efficiency).

Existing algorithms for online fair division assume a small number of item types with a sufficiently large number of copies (Yamada et al., 2024), which allows a learner to compute a good utility estimation using the observed noisy utilities for previous allocations. These utility estimates are then used to select which agent should receive the item, allocating it in a way that maintains a desired balance between ***fairness*** (i.e., keeping the desired level of utilities across the agents) and ***efficiency*** (i.e., maximizing the overall total utility or social welfare) (Sinclair et al., 2022).

Many real-life applications involve a ***large number of items with only a few copies each***. Having a large number of items with only a few copies of each item makes it impossible to estimate the utility for all item-agent pairs, especially using noisy utility observations. For example, as illustrated in Fig. 1, consider an online food delivery platform that wants to recommend restaurants (agents) to its users (items) while balancing between fairly recommending restaurants to

---

[1]Singapore-MIT Alliance for Research and Technology Centre, Singapore [2]Department of Mathematical Sciences, Indian Institute of Technology (BHU), Varanasi, India [3]Graduate School of ISEE, Kyushu University, Japan [4]Department of Computer Science, National University of Singapore, Singapore. Correspondence to: Arun Verma <arun.verma@smart.mit.edu>.

*Proceedings of the 43rd International Conference on Machine Learning*, Seoul, South Korea. PMLR 306, 2026. Copyright 2026 by the author(s).

accommodate their competing interests (fairness) and maximizing its own profit (efficiency). Similar scenarios also arise while recommending the cab to users by ride-hailing platforms, e-commerce platforms choosing top sellers to buyers, resource allocation (Lan et al., 2010; Verma et al., 2019; Verma & Hanawal, 2020), online advertisement (Li et al., 2019), allocating tasks among crowd-sourced workers (Patil et al., 2021; Yamada et al., 2024), providing humanitarian aid post-natural disaster (Yamada et al., 2024), among many others (Walsh, 2011; Mehta et al., 2013; Aleksandrov & Walsh, 2020). Therefore, this problem raises a natural question: ***How can we design an algorithm for online fair division problems having a large number of items with only a few copies for each item?***

The closest works to ours are (Bhattacharya et al., 2024; Procaccia et al., 2024; Yamada et al., 2024; Schiffer & Zhang, 2025), but these works assume that the online fair division problem involves a small number of items and agents, each with a sufficiently large number of copies of each item to ensure good utility estimation for all item-agent pairs. In contrast, we consider an online fair division setting where the number of items can be large with only a few copies for each item, thus considering a more *general problem setting* than in existing works, making existing algorithms (Bhattacharya et al., 2024; Procaccia et al., 2024; Yamada et al., 2024; Schiffer & Zhang, 2025) ineffective in our setting, *as accurately estimating utility for all item-agent pairs is difficult or even infeasible due to the limited copies of each item*. Designing online fair division algorithms for settings with many items and limited copies introduces the following key challenges, which we address using novel techniques.

① **Utility estimation for all item-agent pairs.** The first challenge we face in designing an algorithm for our setting is how to estimate the utility of any item-agent pair using observed stochastic utilities from previous allocations, so that these estimates can be used to select the best agent to allocate an item in the subsequent round. A natural solution is to assume a correlation structure between item-agent features and their corresponding utility, which can be modeled by parameterizing the expected utility as an unknown function of item-agent features.

② **Balancing fairness and efficiency in each round.** The next challenge we face is that, in many real-life applications, agents may also be concerned about fairness in each round of the allocation process (Sim et al., 2021; Benadè et al., 2023), for example, in an online platform, if service providers (agents) do not get an adequate number of users (items), they may choose to leave the platform, or even worse, may join another competing platform. Being fair in each round leads to the next challenge: finding how well the items' allocation to an agent maintains the desired balance between fairness and efficiency. To address this challenge,

we introduce the notion of *goodness function* that measures how well an item allocation to an agent maintains the desired balance between fairness and efficiency (a larger goodness function value implies a better allocation). Thus, the goal is to ensure that the algorithm allocates each item to the agent that maximizes the goodness function.

③ **Exploration-exploitation trade-off.** Even when equipped with estimates of the unknown utility and goodness functions, the algorithm must decide which agent should be allocated the given item. Since the utility is only observed for the agent to whom the item is allocated (i.e., item-agent pair), the algorithm has to deal with the exploration-exploitation trade-off (Auer et al., 2002; Slivkins, 2019; Lattimore & Szepesvári, 2020), i.e., choosing the best agent based on the current utility estimate (exploitation) or trying a new agent that may lead to a better utility estimates in the future (exploration). To deal with the exploration-exploitation trade-off, we adapt the contextual bandit algorithms (Li et al., 2010; Chu et al., 2011; Li et al., 2017) to get an optimistic estimate of unknown utility for each item-agent pair, where the reward in contextual bandits corresponds to utility, contexts to items, and arms to agents. The goodness function (e.g., weighted Gini social-evaluation function (Weymark, 1981) or locally monotonically non-decreasing and Lipschitz functions, more details are in Sec. 3 and Sec. 4.3) uses these optimistic utility estimates to allocate the given item to the best agent that maintains the desired balance between fairness and efficiency. To measure the algorithm's performance in our setting, we use the notion of ***fair regret***, which is the sum of the difference between the maximum goodness function value (i.e., selecting the optimal agent) and goodness function value after the algorithm selects an agent for an item, given the past utilities collected by each agent. The key contributions of this paper are summarized as follows:

- **Online fair division as a contextual bandit.** We study the online fair division problem with noisy, bandit utility feedback and propose algorithms explicitly designed to satisfy the fairness and efficiency trade-off, as outlined in Sec. 4. Furthermore, our algorithmic approach remains effective in settings where only a few copies of each item are available (e.g., online platforms), thereby extending prior work on noisy feedback to more general scenarios.

- **Fairness-efficiency trade-off.** We prove that our proposed algorithms achieve a sub-linear *fair regret* upper bound when the goodness function is the *weighted Gini social evaluation function* (Theorem 1 and Theorem 2), then extend this result to a class of goodness functions that are locally monotonically non-decreasing and locally Lipschitz continuous (Theorem 3 in Sec. 4.3).

- **Empirical results.** Our experiments in Sec. 5 corroborate our theoretical results and validate the performance of our proposed algorithms under different problem settings.

## 1.1. Related Work

In this section, we focus on the most relevant work on online fair division and fair multi-armed bandits, while related work on fair division is discussed in Sec. A.

**Online fair division.** The online fair division problem involves situations in which items arrive sequentially and must be immediately and irrevocably allocated to one of the agents. The nature of items can be different: either divisible (Walsh, 2011) or indivisible (Aleksandrov & Walsh, 2017; Procaccia et al., 2024; Yamada et al., 2024), homogeneous/heterogeneous (Walsh, 2011; Kash et al., 2014), multiple copies of items (Procaccia et al., 2024; Yamada et al., 2024), and agents receiving more than one item (Aleksandrov et al., 2015). In a static setting with all items available upfront, this problem can be formulated as the Eisenberg-Gale (EG) convex program (Eisenberg & Gale, 1959; Jain & Vazirani, 2010). In online settings, envy-freeness (EF) is incompatible with Pareto optimality (PO), even with divisible items. Some works relax EF with approximate solutions, such as relaxed EF (Kash et al., 2014), stochastic approximation for the EG program (Bateni et al., 2022), or approximate fairness (Yamada et al., 2024).

**Fairness and efficiency.** Recent works (Sinclair et al., 2022; Benadè et al., 2023) have examined the trade-off between fairness and efficiency in online allocation problems. Specifically, Benadè et al. (2023) assume that the utilities of all agents for an item are known upon their arrival (noiseless utility). In the online fair allocation problem with partial information, where the algorithm observes ordinal rankings rather than cardinal values, it has been shown that EF and approximate PO can coexist (Benadè et al., 2022). In contrast, we consider a setting in which the utility of an item allocation is unknown and can be observed only through noisy feedback, leading to a learning problem. To address this, existing works (Bhattacharya et al., 2024; Procaccia et al., 2024; Yamada et al., 2024; Schiffer & Zhang, 2025) model online fair division as a multi-armed bandits problem, but only consider problems having a small number of items and agents with a sufficiently large number of copies of each item to ensure a good utility estimation for all item-agent pairs. We instead consider a general setting in which the number of items can be large. However, only a few copies of each item are available, making existing algorithms ineffective for estimating the underlying utilities due to the limited observed utility samples per agent–item pair.

**Fair multi-armed bandits (MAB).** The fair MAB is a framework where an arm with a higher expected reward is selected with a lower probability than an arm with a lower expected reward (Joseph et al., 2018). The fair policy should sample arms with probability proportional to the value of a merit function of their mean reward (Wang & Joachims, 2021). Many works also assume that preserving fairness

means the probability of selecting each arm should be similar if the two arms have a similar quality distribution (Liu et al., 2017; Chen et al., 2021). Fairness is also considered in the linear contextual bandits setting (Gillen et al., 2018; Wu et al., 2023; Schumann et al., 2019; Wang et al., 2021; Grazzi et al., 2022), where an unknown similarity metric imposes individual fairness constraints. Some fair MAB variants seek to optimize the cumulative reward while also ensuring that, at each round, each arm is pulled at least a specified fraction of the time (Li et al., 2019; Chen et al., 2020; Claure et al., 2020; Patil et al., 2021). The work of Hossain et al. (2021) considers a multi-agent setting: the goal is to find a fair distribution of arms among the agents. Recently, Barman et al. (2023) studied Nash regret in stochastic multi-armed bandits with bounded rewards and provided optimal regret guarantees, while Sawarni et al. (2023) extended this analysis to linear bandits.

## 2. Problem setting

**Online fair division.** We consider an online fair division problem involving $N$ agents, in which a learner (or central planner) observes an indivisible item at each round and must allocate it to one of the agents while satisfying a desired level of fairness and efficiency trade-off to the greatest extent possible. We denote the set of agents by $\mathcal{N}$ and the set of indivisible items by $\mathcal{M}$. In our problem, we may have a large number of items with only a few copies each, and the learner has no prior knowledge of their future arrival. At the start of the round $t$, the environment selects an item $m_t \in \mathcal{M}$ (which is drawn from an unknown probability distribution $\nu$), and then the learner observes and allocates the item $m_t$ to an agent $n_t \in \mathcal{N}$. Let $x_{t,n_t} = \varphi(m_t, n_t)$, where $\varphi : \mathcal{M} \times \mathcal{N} \to \mathbb{R}^d$ is a feature map. After that allocation, the learner observes a stochastic utility collected by the agent, denoted by $y_t \doteq f(x_{t,n_t}) + \varepsilon_t$, where $y_t \in \mathbb{R}$, $f : \mathbb{R}^d \to \mathbb{R}^+$ is an unknown utility function, and $\varepsilon_t$ is a $R$-sub-Gaussian noise, i.e., $\forall \lambda \in \mathbb{R}$, $\mathbb{E}\left[e^{\lambda \varepsilon_t} \mid \{x_{s,n_s}, \varepsilon_s\}_{s=1}^{t-1}, x_{t,n_t}\right] \leq \exp\left(\lambda^2 R^2 / 2\right)$.

**Allocation quality measure.** Let $U_{t,n}$ be the cumulative observed utility collected by agent $n$ at the beginning of the round $t$, where $U_{t,n} \doteq \sum_{s=1}^{t-1} y_s \mathbb{1}(n_s = n)$ and $\mathbb{1}(n_s = n)$ denotes the indicator function. We denote the vector of all agents' total utility by $\mathrm{TU}_t$, where $\mathrm{TU}_t \doteq (U_{t,n})_{n \in \mathcal{N}}$. We assume a goodness function G exists that incorporates the desired level needed between fairness and efficiency. This goodness function measures how well the item allocation to an agent maintains the desired balance between fairness and efficiency (a larger value indicates a better allocation). We discuss the goodness function in Sec. 3 and further choices in Sec. F. For the given total utilities of agents, the value of the goodness function G after allocating the item $m_t$ to an agent $n$ is denoted by G $(\mathrm{TU}_{t,n})$. The learner aims to

allocate the item to an agent that maximizes the value of the goodness function or satisfies the given fairness and efficiency trade-off to the greatest extent possible. Here, $\text{TU}_{t,n}$ is the same as $\text{TU}_t$ except the total utility of $n$-th agent is $U_{t,n} + f(x_{t,n})$.

**Performance measure of allocation policy.** Let $n_t^\star$ denote the optimal agent for item $m_t$ having the maximum goodness function value, i.e., $n_t^\star = \operatorname{argmax}_{n \in \mathcal{N}} [\text{G}(\text{TU}_{t,n})]$. Since the utility function $f$ is unknown, we cannot directly compute the optimal agent $n_t^\star$ for item $m_t$. To overcome this, we sequentially estimate the utility function $f$ using the historical information of the stochastic utility observed for the item-agent pairs and then use the estimated utility function to allocate an agent ($n_t$) for the item $m_t$. After allocating item to an agent $n_t$, the learner incurs a penalty (or *instantaneous fair regret*) $r_t$, where $r_t = \text{G}\left(\text{TU}_{t,n_t^\star}\right) - \text{G}\left(\text{TU}_{t,n_t}\right)$. We use this penalty as a performance measure because our goal is to achieve a fair allocation at each round, motivated by practical applications in which the learner must satisfy a desired fairness-efficiency trade-off in each item allocation.

We aim to learn a sequential policy that selects agents for items to minimize the total penalty for not assigning each item to the optimal agent, referred to as *cumulative fair regret*. Specifically, the cumulative fair regret (*regret* for brevity henceforth) of a sequential policy $\pi$ that selects agent $n_t$ to receive item $m_t$ at round $t$ over $T$ rounds is

$$\mathfrak{R}_T(\pi) \doteq \sum_{t=1}^{T} r_t = \sum_{t=1}^{T} \left[ \text{G}\left(\text{TU}_{t,n_t^\star}\right) - \text{G}\left(\text{TU}_{t,n_t}\right) \right]. \quad (1)$$

A policy $\pi$ is a good policy if it has sub-linear regret, i.e., $\lim_{T \to \infty} \mathfrak{R}_T(\pi)/T = 0$. This implies that the policy $\pi$ will eventually start allocating items to the optimal agent.

Note that our regret definition differs from standard contextual bandit algorithms, as the value of the goodness function depends not only on the expected utility in the current round (like in contextual bandits) but also on the utilities collected by each agent in the past (history-dependent). This ***dependence on history makes regret analysis challenging for any arbitrary goodness function***. A similar notion of fair regret has also been used in prior work on fair allocation in an online setting, see regret defined in Eq. 3 of Sim et al. (2021). We provide a detailed discussion of the above regret definition and its different aspects in Sec. B.

## 3. Goodness Function Balancing Fairness and Efficiency

In the following, we define common performance measures for fairness and efficiency in algorithmic game theory and economics. For brevity, let $U_n$ be the utility of agent $n$.

- **Utilitarian Social Welfare** (Feldman & Serrano, 2006).

Utilitarian Social Welfare (USW) is defined as the sum of utilities of all agents, i.e., $\sum_{n \in \mathcal{N}} U_n$. Maximizing USW maximizes the total utility across all agents. We refer to this as an efficiency measure.

- **Egalitarian Social Welfare** (Feldman & Serrano, 2006). Egalitarian Social Welfare (ESW) is the minimum utility across all agents, i.e., $\min_{n \in \mathcal{N}} U_n$. It is a notion of fairness where the designer aims to maximize the utility of less happy agents to obtain a fair outcome.

- **Nash Social Welfare** (Nash et al., 1950). Nash Social Welfare (NSW) is defined as the geometric mean of agents' utilities, i.e., $\left[\prod_{n \in \mathcal{N}} U_n\right]^{\frac{1}{|\mathcal{N}|}}$. Maximizing NSW gets an approximately fair (i.e., envy-free allocation up to one item) and efficient allocation that achieves a balance between USW and ESW.

**Goodness function.** Since the goodness function is a performance indicator of the item allocation to an agent, it is used to allocate the items to agents in a way that maintains the desired balance between fairness and efficiency trade-off. For instance, when a learner aims to maximize social welfare, the item is allocated to an agent with the highest utility. Such an allocation scheme achieves efficiency but sacrifices fairness. One way to obtain fairness is to apply a smaller weight factor to agents with higher cumulative utility and a larger weight factor to those with lower cumulative utility. Therefore, we must consider an appropriate goodness function G such that optimizing G corresponds to a) maximizing efficiency, i.e., maximizing the individual agent's cumulative utility, and b) reducing the utility disparity among the agents, which ensures fairness. Our regret, as defined in Eq. (1), is formulated for a general goodness function G. We first only consider the goodness function to be the *weighted Gini social-evaluation function* (Weymark, 1981), which measures the weighted fair distribution of utility while balancing the trade-off between fairness and efficiency. To define this function, we introduce a *non-negative and non-increasing weight vector* $\boldsymbol{w}_{\mathcal{N}} = (w_1, \ldots, w_{|\mathcal{N}|})$, where $0 \le w_n \le 1$, and $\Phi$ sorts utilities in increasing order ($\Phi(\cdot)_n$ denotes the $n$-th smallest element of its argument). The weighted Gini social-evaluation function is

$$\text{G}\left(\text{TU}_{t,n_t}\right) = \sum_{n \in \mathcal{N}} w_n \, \Phi\left(\text{TU}_{t,n_t}\right). \quad (2)$$

We now consider the following cases that coincide with the well-known social welfare functions.

1. If $w_1 = 1$ and $w_n = 0$ for $n \ge 2$, then the weighted Gini social-evaluation function aligns with ESW, i.e., maximizing the minimum utility among agents.

2. If $w_n = 1$ for all $n \in \mathcal{N}$, then the weighted Gini social-evaluation function aligns with USW, i.e., maximizing the goodness function promotes efficiency.

3. If $0 < w_n \leq 1$ for all $n \in \mathcal{N}$, then by appropriately choosing the weights, we can effectively control the trade-off between efficiency and fairness.

**Remark 1.** *In the first two cases, larger disparities among the weights, with heavier weights on agents with lower cumulative utility, improve fairness, while more uniform weights promote efficiency. To control the trade-off with a single parameter instead of $|\mathcal{N}|$ parameters, we can set the weight $w_n = \rho^{n-1}$, for all $n \in \mathcal{N}$, where $0 \leq \rho \leq 1$ is a control parameter. We experimentally study the impact of parameter $\rho$ on fairness and efficiency in Sec. 5.*

We primarily focus on the weighted Gini social-evaluation function because it allows us to interpolate between fairness (ESW) and efficiency (USW) by adjusting a single parameter $\rho$, as discussed in Remark 1. We also extend our results to other goodness functions, such as NSW and log-NSW; further details are provided in Sec. 4.3 and Sec. F.

## 4. Contextual Bandits for Online Fair Division

The online fair division setting we consider can involve a large number of items with only a few copies (even one); hence, it is statistically challenging to obtain accurate utility estimates for all item-agent pairs due to limited noisy utility observations. To overcome this challenge, we assume a correlation between the utility and item-agent features, which can be realized by parameterizing the expected utility so that it depends on an unknown function of the item-agent features. This assumption is common in the contextual bandit literature (Li et al., 2010; Chu et al., 2011; Agrawal & Goyal, 2013; Zhou et al., 2020; Zhang et al., 2021), where the reward (utility) is assumed to be an unknown function of context-arm (item-agent) features.

In this paper, we model online fair division of a large number of items with a few copies as a contextual bandit problem, where the items serve as contexts, the agents as arms, and utility as rewards. *Compared to standard contextual bandit algorithms, our proposed algorithms are explicitly designed to satisfy the fairness and efficiency trade-off.* To bring out our key ideas and results, we first focus on linear utility functions with the goodness function defined in Eq. (2), and later extend our results to non-linear functions in Sec. 4.2 and to other goodness functions in Sec. 4.3.

### 4.1. Linear Utility Function

For brevity, let $\mathcal{X} \subset \mathbb{R}^d$ be the set of all known item-agent feature vectors. At each round $t$, for any item-agent pair $(m_t, n)$, the corresponding feature vector $x_{t,n} \in \mathcal{X}$ is observed by the learner, where $d \geq 1$. After allocating the item $m_t$ to an agent $n_t$, the learner observes stochastic utility $y_t = x_{t,n_t}^\top \theta^\star + \varepsilon_t$, where $\theta^\star \in \mathbb{R}^d$ is the unknown parameter and $\varepsilon_t$ is $R$-sub-Gaussian noise. Let

$M_t \doteq \sum_{s=1}^{t-1} x_{s,n_s} x_{s,n_s}^\top + \lambda I_d$, where $x_{s,n_s}$ is the item-agent features in round $s \leq t$, $\lambda > 0$ is the regularization parameter ensuring $M_t$ is a positive definite matrix, and $I_d$ is the $d \times d$ identity matrix. The weighted $l_2$-norm of vector $m$ with respect to matrix $M$ is denoted by $\|m\|_M$. At the start of round $t$, $\hat{\theta}_t = M_t^{-1} \sum_{s=1}^{t-1} x_{s,n_s} y_s$ is the estimate of the unknown parameter $\theta^\star$.

After obtaining this utility function estimator, the learner must decide which agent to allocate the given item to. Since the utility is only observed for the selected item-agent pair, the learner needs to deal with the exploration-exploitation trade-off (Auer et al., 2002; Lattimore & Szepesvári, 2020), i.e., choosing the best agent based on the current utility estimate (exploitation) or trying a new agent that may lead to a better utility estimator in the future (exploration). The upper confidence bound (UCB) (Li et al., 2010; Chu et al., 2011; Zhou et al., 2020) and Thompson sampling (TS) (Krause & Ong, 2011; Agrawal & Goyal, 2013; Zhang et al., 2021) are widely-used techniques for dealing with the exploration-exploitation trade-off.

---

**OFD-UCB** UCB-based algorithm for online fair division with linear utility function

---

1: **Input:** $\lambda > 0$, Number of agents $N = |\mathcal{N}|$
2: For the first $N$ rounds: allocates items to agents in a round-robin fashion
3: Compute $M_{N+1} = \lambda I_d + \sum_{s=1}^{N} x_{s,n_s} x_{s,n_s}^\top$ and $\hat{\theta}_{N+1} = M_{N+1}^{-1} \sum_{s=1}^{N} x_{s,n_s} y_s$
4: **for** $t = N+1, N+2, \ldots$ **do**
5:     Observe an item $m_t$
6:     Select agent $n_t \in \text{argmax}_{n \in \mathcal{N}} \, \text{G}\left(\text{TU}_{t,n}^{\text{UCB}}\right)$. Break ties randomly
7:     Observe stochastic utility $y_t$ for agent $n_t$
8:     Update $M_{t+1} = M_t + x_{t,n_t} x_{t,n_t}^\top$ and re-estimate $\hat{\theta}_{t+1} = M_{t+1}^{-1} \sum_{s=1}^{t} x_{s,n_s} y_s$
9: **end for**

---

**UCB-based algorithm.** We first propose a UCB-based algorithm, named **OFD-UCB**, that works as follows. In the first $N$ rounds, the learner allocates items to agents in a round-robin fashion to ensure each agent has positive utility. At round $t$, the environment reveals an item $m_t$ to the learner. Before selecting the agent for that item, the learner updates the utility function estimate ($\hat{\theta}_t$) using available historical information before the round $t$ (i.e., $\{x_{s,n_s}, y_s\}_{s=1}^{t-1}$). Then, the utility UCB value for allocating item $m_t$ to an agent $n$ is

$$u_{m_t,n}^{\text{UCB}} = x_{t,n}^\top \hat{\theta}_t + \alpha_t \|x_{t,n}\|_{M_t^{-1}},$$

where $x_{t,n}^\top \hat{\theta}_t$ is the estimated utility for allocating item $m_t$ to agent $n$ and $\alpha_t \|x_{t,n}\|_{M_t^{-1}}$ is the confidence bonus in which $\alpha_t = R\sqrt{d \log\left(\frac{1+(tL^2/\lambda)}{\delta}\right)} + \lambda^{\frac{1}{2}} S$ is a slowly increasing

function in $t$ and the value of $\|x_{t,n}\|_{M_t^{-1}}$ goes to zero as $t$ increases. Here, $S$ and $L$ are defined such that $\|\theta^\star\|_2 \leq S$ and $\|x_{t,n}\|_2 \leq L$ for all $t \geq 1$ and $n \in \mathcal{N}$. Using this optimistic utility estimate of each agent, the algorithm selects an agent by maximizing the optimistic value of the goodness function: $n_t \in \arg\max_{n \in \mathcal{N}} \mathrm{G}\left(\mathrm{TU}_{t,n}^{\mathrm{UCB}}\right)$, where $\mathrm{TU}_{t,n}^{\mathrm{UCB}} = \left(U_{t,a} + u_{m_t,a}^{\mathrm{UCB}}\mathbb{1}(a = n)\right)_{a \in \mathcal{N}}$ in which $U_{t,a}$ is the total utility collected by the agent $a$ before the round $t$.

If there are multiple agents to whom allocating the item gives the same maximum value of the goodness function, then one of these agents is chosen randomly. After allocating item $m_t$ to agent $n_t$, the environment generates a stochastic utility $y_t$. The learner observes the utility $y_t$ and then updates the values of $M_{t+1} = M_t + x_{t,n_t}x_{t,n_t}^\top$ and re-estimates $\hat{\theta}_{t+1} = M_{t+1}^{-1}\sum_{s=1}^{t} x_{s,n_s}y_s$. The same process is repeated to select agents for subsequent items. Our proposed algorithms explicitly enforce a fairness-efficiency trade-off by selecting an agent for each item allocation based on UCB values. Our first result provides a regret upper bound of **OFD-UCB** under a linear utility function.

**Theorem 1.** *Let $\delta \in (0, 1)$, $\lambda > 0$, $\|\theta^\star\|_2 \leq S$, $\|x_{t,n}\|_2 \leq L \ \forall t \geq 1, n \in \mathcal{N}$, noise in utility be the $R$-sub-Gaussian, and the goodness function be the same as defined in Eq. (2) with $w_{\max} = \max_{n \in \mathcal{N}} w_n$. Then, with a probability of at least $1 - \delta$, the regret in $T > 0$ rounds is*

$$\mathfrak{R}_T\left(\textbf{OFD-UCB}\right) \leq 2\alpha_T w_{\max}\sqrt{2dT\log(\lambda + TL^2/d)},$$

*where $\alpha_T = R\sqrt{d\log\left(\frac{1+(TL^2/\lambda)}{\delta}\right)} + \lambda^{\frac{1}{2}}S$.*

**Proof outline.** The key observation is that cumulative regret depends on the instantaneous regret incurred in each round. We can upper bound the instantaneous regret $(r_t)$ incurred in the round $t$ by $2w_{\max}\left\|\hat{\theta}_t - \theta^\star\right\|_{M_t}\|x_{t,n_t}\|_{M_t^{-1}}$. As the instantaneous regret depends on the estimation error of $\theta^\star$ using observed item-agent pairs, i.e., $\left\|\hat{\theta}_t - \theta^\star\right\|_{M_t}$ in the round $t$, we adapt results from linear contextual bandits to our setting to get an upper bound on this estimation error. With this result, the regret upper bound follows by upper-bounding the sum of all instantaneous regret. The detailed proofs of Theorem 1 and proofs of other related results are provided in Sec. C.

**Algorithm based on Thompson sampling.** Due to the empirical superiority of TS-based algorithms over UCB-based bandit algorithms (Chapelle & Li, 2011; Agrawal & Goyal, 2013), we propose a TS-based algorithm, **OFD-TS**, which mirrors **OFD-UCB** except for agent selection (Line 6). To get a TS-based utility estimate, the algorithm first samples a utility function parameter $\tilde{\theta}_t$ from the distribution Normal $\left(\hat{\theta}_t, \beta_t^2 M_t^{-1}\right)$, where Normal denotes the normal

distribution and $\beta_t = R\sqrt{9d\log(t/\delta)}$ (Agrawal & Goyal, 2013). The utility estimate $u_{m_t,n}^{\mathrm{TS}} = x_{t,n}^\top \tilde{\theta}_t$ then replaces $u_{m_t,n}^{\mathrm{UCB}}$ for computing the value of goodness function.

## 4.2. Non-linear Utility Function

We now consider the setting in which the utility function can be non-linear. As shown in Sec. 4.1, the linear contextual bandit algorithm can be used as a subroutine to get the optimistic utility estimate in our contextual online fair division problem. These estimates are then used to compute the goodness function for allocating the observed item to the agent that provides the best balance between fairness and efficiency. We generalize this observation for online fair division problems with a non-linear utility function by using a suitable non-linear contextual bandit algorithm and introduce the notion of *Online Fair Division* (OFD) *Compatible* contextual bandit algorithm.

**Definition 1** (**OFD Compatible Contextual Bandit Algorithm**). *Let $f$ be the unknown utility function, $\mathcal{O}_t$ denote the observations of item-agent pairs at the beginning of round $t$, and $x_{t,n} \in \mathcal{X}$. Then, any contextual bandit algorithm $\mathfrak{A}$ is OFD Compatible if its estimate $f_t^{\mathfrak{A}}$ of the utility function $f$ satisfies, with probability $1 - \delta$: $|f_t^{\mathfrak{A}}(x_{t,n}) - f(x_{t,n})| \leq h(x_{t,n}, \mathcal{O}_t)$, where $h(\cdot, \cdot)$ depends on $x_{t,n}$ and past observations $\mathcal{O}_t$.*

Many contextual bandit algorithms like Lin-UCB (Chu et al., 2011), UCB-GLM (Li et al., 2017), IGP-UCB (Chowdhury & Gopalan, 2017), GP-TS (Chowdhury & Gopalan, 2017), Neural-UCB (Zhou et al., 2020), and Neural-TS (Zhang et al., 2021) are OFD compatible. The value of $h(x_{t,n}, \mathcal{O}_t)$ provides an upper bound on the goodness function of the estimated utility with respect to the true utility function. This value depends on the problem and the choice of contextual bandit algorithm $\mathfrak{A}$ and its associated hyperparameters (e.g., $\delta$; see Tab. 1 in the Appendix for more details). For a given problem, any appropriate OFD compatible contextual bandit algorithm can be used as a subroutine to get optimistic utility estimates for all item-agent pairs. These estimates are used to compute the goodness function value, which is used to select the best agent for allocating the given item.

Let $\mathfrak{A}$ be an OFD contextual bandit algorithm with $|f_t^{\mathfrak{A}}(x_{t,n}) - f(x_{t,n})| \leq h(x_{t,n}, \mathcal{O}_t)$. Then, the agent for allocating the item $m_t$ is selected by maximizing the goodness function: $n_t \in \arg\max_{n \in \mathcal{N}} \mathrm{G}\left(\mathrm{TU}_{t,n}^{\mathfrak{A}}\right)$, where $\mathrm{TU}_{t,n}^{\mathfrak{A}} = \left(U_{t,a} + u_{m_t,a}^{\mathfrak{A}}\mathbb{1}(a = n)\right)_{a \in \mathcal{N}}$ in which $U_{t,a}$ is the total utility collected by agent $a$ thus far and $u_{m_t,a}^{\mathfrak{A}}$ is the optimistic estimate of $f(x_{t,a})$ (e.g., $u_{m_t,a}^{\mathfrak{A}} = f_t^{\mathfrak{A}}(x_{t,a}) + h(x_{t,a}, \mathcal{O}_t)$ for UCB-based contextual bandit algorithms). Note that the assumptions underlying contextual bandit algorithms must also be satisfied in our setting, as they directly influence the performance of our proposed algorithm via the function

$h(x_{t,n_t}, \mathcal{O}_t)$. Next, we provide a regret upper bound for any OFD compatible contextual bandit algorithm that produces optimistic utility estimates for agents given an item.

**Theorem 2.** *Let $\mathfrak{A}$ be an OFD compatible contextual bandit algorithm with $|f_t^{\mathfrak{A}}(x_{t,n}) - f(x_{t,n})| \leq h(x_{t,n}, \mathcal{O}_t)$ and the goodness function be same as defined in Eq. (2) with $w_{\max} = \max_{n \in \mathcal{N}} w_n$. If the assumptions underlying $\mathfrak{A}$ holds, then, with a probability of at least $1 - \delta$, the regret of corresponding OFD algorithm **OFD-$\mathfrak{A}$** in $T$ rounds is*

$$\mathfrak{R}_T\left(\textbf{OFD-}\mathfrak{A}\right) \leq 2w_{\max}\sqrt{T}\sqrt{\sum_{t=1}^{T}\left[h(x_{t,n_t}, \mathcal{O}_t)\right]^2}.$$

**Proof outline.** The proof follows by upper-bounding the sum of instantaneous regrets. Specifically, we first show that instantaneous regret $(r_t)$ incurred in the round $t$ is upper bounded by $2w_{n_t}h(x_{t,n_t}, \mathcal{O}_t)$ (as shown in Sec. C), using the fact that the estimation error is $|f_t^{\mathfrak{A}}(x_{t,n}) - f(x_{t,n})| \leq h(x_{t,n}, \mathcal{O}_t)$ for an item-agent pair $x_{t,n}$ when using an OFD compatible contextual bandit algorithm $\mathfrak{A}$. Thus, the regret of **OFD-$\mathfrak{A}$** directly depends on $\mathfrak{A}$ through $h(x_{t,n_t}, \mathcal{O}_t)$, resulting in the same order of regret as that of $\mathfrak{A}$, since $w_{\max}$ is a scale-free constant independent of $N$. Note that **OFD-$\mathfrak{A}$** incurs linear regret if the underlying bandit algorithm $\mathfrak{A}$ itself suffers linear regret.

### 4.3. Locally Monotonically Non-decreasing and Lipschitz Goodness Functions

We have also derived regret upper bounds for goodness functions that satisfy the local monotonicity and local Lipschitz continuity properties.

**Definition 2** (**Locally monotonically non-decreasing and locally Lipschitz continuous function**)**.** *Let $TU_t \in \mathbb{R}^N$ denote the utility vector at round $t$, and let $TU_{t,-n}$ denote the vector of utilities excluding the $n$-th agent. A goodness function $G : \mathbb{R}^N \to \mathbb{R}$ is said to satisfy the following properties for each element-wise $n \in [N]$: (i) **locally monotonically non-decreasing** for any $u \in \mathbb{R}^+$ if*

$$G\left(U_{t,n}, TU_{t,-n}\right) \leq G\left(U_{t,n} + u, TU_{t,-n}\right), \textit{ and}$$

*(ii) **locally Lipschitz continuous** if, for a constant $c_n > 0$ such that for any $U_{t,n}, U_{t,n}' \in \mathbb{R}$,*

$$|G\left(U_{t,n}, TU_{t,-n}\right) - G\left(U_{t,n}', TU_{t,-n}\right)| \leq c_n|U_{t,n} - U_{t,n}'|.$$

Next, we present an upper bound on the regret for goodness functions that are locally monotonically non-decreasing and locally Lipschitz continuous, such as NSW and log-NSW.

**Theorem 3.** *Let $\mathfrak{A}$ be an OFD compatible contextual bandit algorithm with $|f_t^{\mathfrak{A}}(x_{t,n}) - f(x_{t,n})| \leq h(x_{t,n}, \mathcal{O}_t)$ and the goodness function $G$ is locally monotonically non-decreasing and locally Lipschitz continuous, with $c_{\max} =$*

$\max_{n \in \mathcal{N}} c_n$. *If the assumptions used in $\mathfrak{A}$ holds, then, with a probability of at least $1 - \delta$, the regret of corresponding OFD algorithm **OFD-$\mathfrak{A}$** in $T$ rounds is*

$$\mathfrak{R}_T\left(\textbf{OFD-}\mathfrak{A}, \text{G}\right) \leq 2c_{\max}\sqrt{T}\sqrt{\sum_{t=1}^{T}\left[h(x_{t,n_t}, \mathcal{O}_t)\right]^2}.$$

**Proof outline.** The instantaneous regret $(r_t)$ is upper bounded by $2c_{n_t}h(x_{t,n_t}, \mathcal{O}_t)$ (as shown in Sec. C). This result uses the following three facts: (i) the locally monotonically non-decreasing property, (ii) the locally Lipschitz property of the goodness function, and (iii) the estimation error bound $|f_t^{\mathfrak{A}}(x_{t,n}) - f(x_{t,n})| \leq h(x_{t,n}, \mathcal{O}_t)$ for the contextual bandit algorithm $\mathfrak{A}$.

## 5. Experiments

In this section, we aim to corroborate our theoretical results and empirically demonstrate the performance of our proposed algorithms in different online fair allocation problems. We repeat all our experiments 20 times and report regret (see Eq. (1)) with 95% confidence intervals (the vertical lines on each curve indicate the confidence intervals). To validate the different performance aspects of our proposed algorithms, we have used synthetic problem instances (commonly used in the bandit literature) with the following details.

**Experiment setting.** We use a $d_m$-dimensional space to generate the sample features of each item, where item $m_t$ is represented by $m_t = (m_{t,1}, \ldots, m_{t,d_m})$ for $t \geq 1$. Similarly, we use a $d_n$-dimensional feature space to generate the sample features of each agent. Specifically, each agent $n_t \in \mathcal{N}$ is represented by a feature vector $n_t = (n_{t,1}, \ldots, n_{t,d_n})$. The value of the $i$-th feature $m_{t,i}$ (or $n_{t,i}$) is sampled uniformly at random from $(0, 10)$. Note that the set of agents remains the same across the rounds, whereas items are independently sampled from the $d_m$-dimensional space. To construct the item-agent feature vectors for item $m_t$ in the round $t$, we concatenate the item features $m_t$ with the feature vector of each agent. For item $m_t$ and agent $n$, the concatenated feature vector $x_{t,n}$ is a $d$-dimensional vector with $d = d_m + d_n$. We select a $d$-dimensional vector $\theta^\star$ by sampling uniformly at random from $(0, 10)^d$ and normalizing it to have unit $l_2$-norm. In all experiments, we use $\lambda = 0.01$, $R = 0.1$, $\delta = 0.05$, and $d_m = d_n$.

**Regret comparison with baselines.** To the best of our knowledge, this paper is the first work to model online fair division of numerous items with few copies using a contextual bandit framework. We compare the regret of proposed algorithms with two baselines: OFD-Uniform and OFD-Greedy. OFD-Uniform selects an agent uniformly at random to allocate the item. In contrast, OFD-Greedy uses $\varepsilon$-Greedy contextual algorithm, which behaves the same

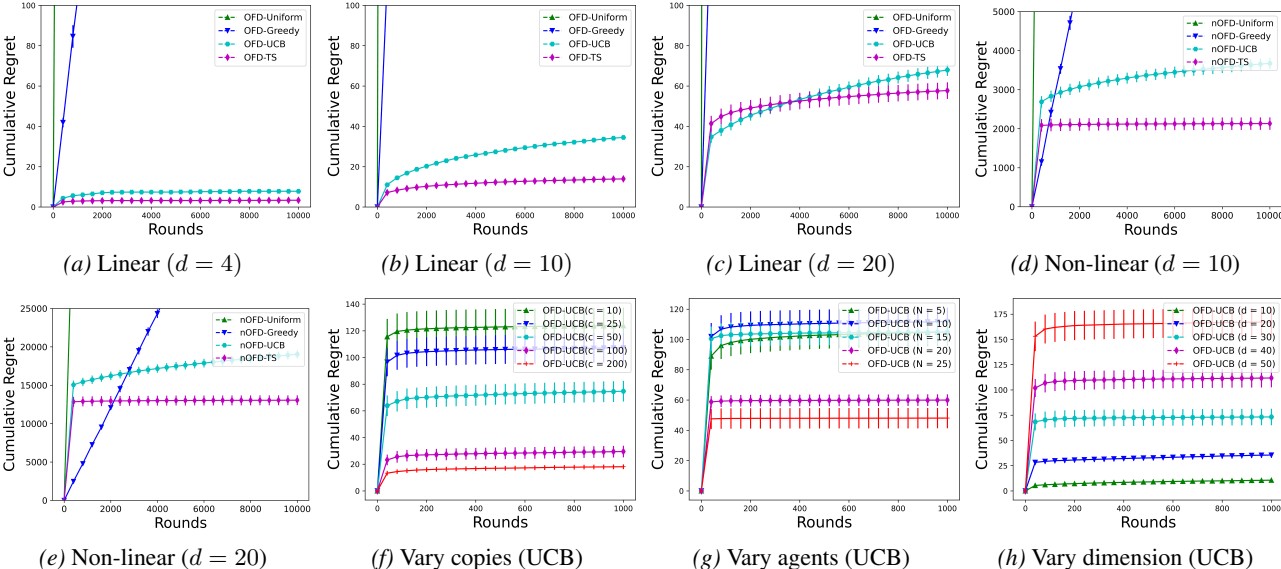

*Figure 2.* Comparing cumulative regret of our proposed online fair division algorithms with baseline algorithms ((**Fig. 2a**-**Fig. 2c**): for linear utility function and (**Fig. 2d**-**Fig. 2e**) for non-linear utility function). (**Fig. 2f** - **Fig. 2h**): Cumulative regret of **OFD-UCB** vs. different number of copies for each item ($c$), number of agents ($N$), and dimension of item-agent feature vector ($d$).

as **OFD-UCB** except it has no confidence term (i.e., setting $\alpha_t = 0$) and selects an agent uniformly at random with probability 0.1 in each round, else otherwise select agent greedily. For experiments with the linear utility (i.e., $f(x_{t,n}) = x_{t,n}^\top \theta^\star$), we use 10000 items, 10 agents, and $\rho = 0.85$ control parameter for the goodness function.

We use three problems that share the same setting, differing only in the feature dimensions $d_m = d_n = \{2, 5, 10\}$, resulting in $d = \{4, 10, 20\}$. We use a polynomial kernel of degree 2 for a non-linear utility function to transform the item-agent feature vectors to introduce non-linearity; more details are in Sec. D. For experiments involving non-linear utility functions, we use 1000 items, 10 agents, $\rho = 0.85$, and $d_m = d_n = \{5, 10\}$, resulting in $d = \{10, 20\}$. As expected, our algorithms based on UCB and TS-based contextual linear bandit algorithms outperform both baselines as shown in Fig. 2a–2e on different problem instances of linear utility and non-linear utility functions (only varying the dimension $d$ while keeping remaining parameters unchanged). Note that we limit the y-axis range to better highlight the sublinear regret of our algorithm. Furthermore, as expected, the TS-based algorithm consistently outperforms the UCB-based algorithm, reflecting the superior empirical performance of TS-based algorithms.

**Regret vs. number of copies for each item.** Having multiple copies of each item makes it easier to estimate the unknown utility function, thereby reducing regret. To verify this effect, we analyze how regret varies with the number of copies per item ($c$). In our experiments, we use a linear utility function ($f(x_{t,n}) = x_{t,n}^\top \theta^\star$), a total of $1000/c$

items, $N = 10$, $\rho = 0.85$, $d = 40$, and vary $c$ from the set $\{10, 25, 50, 100, 200\}$. As expected, the regret of our UCB-based algorithm decreases with more copies per item, as shown in Fig. 2f.

**Regret vs. number of agents** ($N$) **and dimension** ($d$). The number of agents ($N$) and the dimension of the item-agent feature vector ($d$) in the online fair division problem control the difficulty. As their values increase, the problem becomes more difficult, making it harder to allocate the item to the best agent. We want to verify this by observing how the regret of our proposed algorithms changes as $N$ and $d$ vary in the online fair division problem. To see this in our experiments, we use the linear utility function (i.e., $f(x_{t,n}) = x_{t,n}^\top \theta^\star$), 1000 items, $N = 10$ when varying dimension, $\rho = 1$, $d = 40$ while varying the number of agents. As shown in Fig. 2g, the regret bound of our UCB-based algorithm increases as we increase the number of agents, i.e., $N = \{5, 10, 15, 20, 25\}$. We also observe the same trend when increasing the dimension of the item-agent feature vector from $d = \{10, 20, 30, 40, 50\}$ as shown in Fig. 2h.

**Total utility, Gini coefficient, and ratio of minimum utility to total utility vs.** $\rho$**.** To measure the fairness of item allocation among agents using the Gini coefficient (Schutz, 1951; Dorfman, 1979) (lower is better) and the ratio of minimum utility to total utility (increasing this is the same as maximizing max-min utility, hence having a higher value is better). We can use total utility (USW) as a measure of efficiency (higher total utility is better). Since the value of $\rho$ controls the balance between fairness and efficiency, we want to see how it changes total utility, the Gini coefficient,

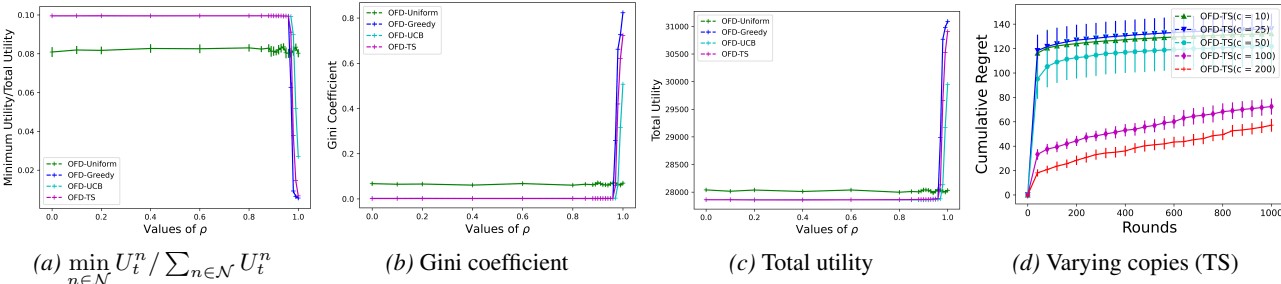

*(a)* $\min_{n \in \mathcal{N}} U_t^n / \sum_{n \in \mathcal{N}} U_t^n$     *(b)* Gini coefficient     *(c)* Total utility     *(d)* Varying copies (TS)

*Figure 3.* **Fig. 3a-Fig. 3c:** Fairness and efficiency measures (ratio of minimum utility to total utility, Gini coefficient, and total utility) as a function of control parameter ($\rho$). **Fig. 3d:** Cumulative regret of **OFD-TS** vs. different item's copies ($c$).

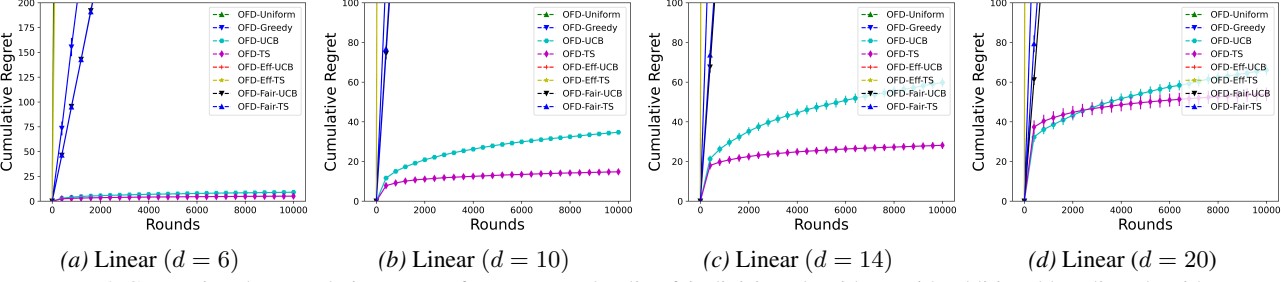

*(a)* Linear ($d = 6$)     *(b)* Linear ($d = 10$)     *(c)* Linear ($d = 14$)     *(d)* Linear ($d = 20$)

*Figure 4.* Comparing the cumulative regret of our proposed online fair division algorithms with additional baseline algorithms.

and the ratio of minimum utility to total utility. To see this, we use the linear utility function (i.e., $f(x_{t,n}) = x_{t,n}^\top \theta^\star$), 1000 items, $N = 10$, $d = 40$, and vary value in $\rho$ in set $\{0, 0.1, 0.2, 0.4, 0.6, 0.8, 0.85, 0.88, 0.89, 0.9, 0.91, 0.92, 0.93, 0.94, 0.95, 0.96, 0.97, 0.98, 0.99, 1.0\}$.

For each value of $\rho$, we note the total utility, Gini coefficient, and ratio of minimum utility to total utility after allocating all items. We repeat our experiments 20 times and report the average value of observed results in Fig. 3a-3c. As $\rho$ increases, efficiency is preferred over fairness. As expected, total utility increases (Fig. 3c), the value of the Gini coefficient increases (Fig. 3b), and the ratio of minimum utility to total utility decreases (Fig. 3a). As expected, OFD-Uniform performs worse. Though OFD-Greedy behaves the same as UCB- and TS-based counterparts, it suffers from higher regret as shown in Fig. 3a-3c.

**Regret comparison with additional baselines.** We compare the regret of the proposed algorithms against four additional baselines beyond those used in the main paper: OFD-Eff-UCB/TS and OFD-Fair-UCB/TS. The OFD-Eff-UCB/TS algorithms select agents to maximize the Utilitarian Social Welfare (USW). In contrast, the OFD-Fair-UCB/TS algorithms select agents to maximize the Egalitarian Social Welfare (ESW), using UCB or TS strategies. We run all experiments with the linear utility (i.e., $f(x_{t,n}) = x_{t,n}^\top \theta^\star$). We use 10000 items, 10 agents, and $\rho = 0.85$ as the control parameter for the goodness function. We use four different problems with the same setting except $d_m = d_n = \{3, 5, 7, 10\}$, resulting in $d = \{6, 10, 14, 20\}$. As expected, our UCB- and TS-based contextual linear

bandit algorithms outperform all baselines, as shown in Fig. 4a-4d across different problem instances of linear utility (varying only the dimension $d$ while keeping the remaining parameters unchanged). Note that we set a limit on the y-axis to highlight the sub-linear regret of our algorithm. Furthermore, as expected, we also observe that the TS-based algorithm performs better than the UCB-based algorithm.

Due to space constraints, we defer additional experimental results to Sec. D. These include results from the TS-based algorithm and experiments with varying values of $\rho$.

## 6. Conclusion

This paper studies a novel variant of the online fair division problem, involving a large number of items with only a few copies each. As items arrive sequentially, a learner must irrevocably allocate each indivisible item to one of the agents while satisfying the desired fairness-efficiency trade-off. We then propose algorithms with sub-linear regret guarantees that are explicitly designed for the aforementioned trade-off. To handle the exploration-exploitation trade-off, these algorithms adapt contextual bandit algorithms to online fair division, obtaining optimistic utility estimates for each item-agent pair and using them for item allocation to agents. Our experimental results further validate the performance of the proposed algorithms. An important direction for future work is to generalize the framework to more expressive valuation functions beyond the current setting, including other fairness notions such as envy-freeness and proportionality. Another promising avenue is to study fairness across multiple item types in online fair division with unknown utility functions.

## Acknowledgements

This research is supported by the National Research Foundation (NRF), Prime Minister's Office, Singapore under its Campus for Research Excellence and Technological Enterprise (CREATE) programme. The Mens, Manus, and Machina (M3S) is an interdisciplinary research group (IRG) of the Singapore MIT Alliance for Research and Technology (SMART) centre. The author affiliated with Kyushu University is partially supported by JST ERATO Grant Number JPMJER2301.

## Impact Statement

This work is primarily theoretical, focusing on the design and analysis of algorithms. The proposed methods do not directly involve human subjects, personal data, or real-world deployments. While the algorithm could potentially be applied in systems that interact with users, we emphasize that ethical considerations, such as fairness, privacy, and informed consent, must be addressed in practical deployments. Our primary goal is to advance the theoretical understanding of efficiency-fairness tradeoff in contextual bandit algorithms, and we do not anticipate any immediate negative societal impacts.

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

## A. Additional Related Work

Due to space constraints in the main paper, we will now highlight relevant work in fair division and fair multi-armed bandits.

**Fair division.** The fair division is a fundamental problem (Steinhaus, 1948; Dubins & Spanier, 1961) in which a designer allocates resources to agents with diverse preferences. There are two popular notions of 'fairness' used in fair division: envy-freeness (EF) (Varian, 1974) (every agent prefers his or her allocation most) and proportional fairness (Steinhaus, 1948) (each agent is guaranteed her proportional share in terms of total utility independent of other agents). Many classical studies on fair division focus on divisible resources (Steinhaus, 1948; Dubins & Spanier, 1961; Varian, 1974), which are later extended to indivisible items (Brams & Taylor, 1996; Moulin, 2004; Budish, 2011) due to their many practical applications. However, an envy-free allocation may not exist, e.g., for two agents and one item. Therefore, the appropriate relaxations of EF and proportionality, such as envy-freeness up to one good (EF1) (Moulin, 2004), envy-freeness up to any good (EFX) (Caragiannis et al., 2019), and maximin share fairness (MMS) (Budish, 2011), are used in practice. We refer the readers to (Amanatidis et al., 2022) for a survey on fair division.

## B. Discussion about Regret defined in Eq. (1)

**Goodness function.** Since the utility function $f$ (and hence $G(\cdot)$) is unknown, our proposed algorithms need to estimate the agent's utility to compute the goodness function's value before allocating a given item. To do this, we can use a suitable contextual bandit algorithm that provides an optimistic utility estimate for each item-agent pair. The value of goodness function $G$ for allocating the item $m_t$ to an agent $n$ is denoted by $G(U_{t,n})$, where $U_{t,n}$ represents the vector of all agents' total utility at the beginning of round $t$, except that the total utility of $n$-th agent is $U_{t,n} + f(x_{t,n})$, i.e., optimistic estimate of utility if item $m_t$ will be allocated to an agent $n$.

**Motivation for regret defined in Eq. (1).** Our goal is fair allocation each round, motivated by practical applications where learners ensure fairness in item distribution (Procaccia et al., 2024; Sim et al., 2021; Clerici et al., 2024). For example, on an online platform, service providers (agents) may leave or switch to competitors if they receive too few users. Thus, our algorithms optimize the goodness function each round to balance fairness and efficiency. By design, this definition of regret satisfies the Markovian property; that is, the item allocation to an agent depends only on the current total utility (sum of the observed noisy utilities, i.e., $y_t$'s of the agents and an optimistic estimate of their utility. Furthermore, the oracle uses the function $f$ to compute the goodness function $G$ for each agent before allocating an item to the best agent. Since $y_t$ represents the observed noisy utility after allocation in round $t$, the oracle does not know the observation noise $\varepsilon_t$. As a result, Oracle can not use $U_{t,n} + f(x_{t,n}) + \varepsilon_t$ to compute the goodness function $G$ for the $n$-th agent. Furthermore, it is a standard practice to ignore noise $\varepsilon_t$ when computing the optimal arm in contextual bandit algorithms.

**Regret with respect to *Oracle*'s utilities.** Note that the goodness function used in the regret definition from Eq. (1) is based on the total utilities collected by agents under the current policy. The oracle policy in this context uses the current allocation history to determine the optimal agent to select, based on the available allocations at each round.

Since our regret definition depends on agents' past allocations, the oracle policy also influences the total utility each agent accrues. To better reflect this dependency, we define a stronger oracle policy that accounts for the total utilities agents would have accumulated if they had always followed the oracle. Let $\mathrm{TU}_{t,n}^\star$ denote the total utility of agent $n$ up to time $t$ under the oracle policy. Then, we define the regret as:

$$\mathfrak{R}_T(\pi, \star) \doteq \sum_{t=1}^T \left[ \mathrm{G}\left(\mathrm{TU}_{t,n_t^\star}^\star\right) - \mathrm{G}\left(\mathrm{TU}_{t,n_t}\right) \right]. \tag{3}$$

This refined regret definition introduces an additional term in the regret bound. As shown at the beginning of the proof of Theorem 3, the third inequality no longer benefits from a cancellation of terms. Instead, an extra component appears in the instantaneous regret at round $t$, i.e., $c_{n_t} \left| \mathrm{TU}_{t,n_t}^\star - \mathrm{TU}_{t,n_t} \right|$. Let $d_t := c_{n_t} \left| \mathrm{TU}_{t,n_t}^\star - \mathrm{TU}_{t,n_t} \right|$. Then the cumulative additional regret over $T$ rounds is given by: $D_T = \sum_{t=1}^T d_t$. Bounding $D_T$ rigorously is left for future work.

## C. Regret Analysis.

We first state the following result that we will use in the proof of Theorem 1.

**Lemma 1** (Theorem 2 of (Abbasi-Yadkori et al., 2011)). *Let* $\delta \in (0, 1)$, $\lambda > 0$, $\hat{\theta}_t = M_t^{-1} \sum_{s=1}^t x_{s,n} y_s$, $\|\theta^\star\|_2 \leq S$, *and*

$\|x_{t,n}\|_2 \le L \ \forall t \ge 1, n \in \mathcal{N}$. *Then, with probability $1 - \delta$,*

$$\left\| \hat{\theta}_t - \theta^\star \right\|_{M_t} \le \left( R\sqrt{d \log\left( \frac{1 + (tL^2/\lambda)}{\delta} \right)} + \lambda^{\frac{1}{2}} S \right) = \alpha_t.$$

We next state the following result that we will use to prove our regret upper bounds for Theorem 1 and Theorem 2.

**Lemma 2** (Lemma 1 of (Weymark, 1981))**.** *Let $\boldsymbol{w} = (w_1, w_2, \ldots, w_N)$ and $\boldsymbol{u} = (u_1, u_2, \ldots, u_N)$ be opposite ordered. If $TU = \{\boldsymbol{u}^1, \boldsymbol{u}^2, \ldots, \boldsymbol{u}^p\}$ is the set of all permutations of $\boldsymbol{u}$, then*

$$\sum_{n=1}^N w_n u_n \le \sum_{n=1}^N w_n u_n^j, \quad \text{for } j = 1, 2, \ldots, p.$$

**Lemma 3** (Lemma 1 of (Sim et al., 2021))**.** *Let $w_1 > \ldots > w_N > 0$ and $G(TU) = \sum_{n \in \mathcal{N}} w_n \, \Phi(TU)$, where $TU$ is an $N$-dimensional utility vector and $\Phi$ returns the $n$-th smallest element of $TU$. For any two utility vectors $TU^a$ and $TU^b$, if there exists $i$ such that $U_{i,a} > U_{i,b}$ and $\forall n \in \mathcal{N} \setminus \{i\}$, $U_{n,a} = U_{n,b}$, then $G(TU^a) > G(TU^b)$.*

When all weights are equal (i.e., $\rho = 1$, corresponding to USW), the conclusion of Lemma 3 holds trivially because USW is additively separable and strictly increasing in any individual agent's utility.

The regret analysis of our proposed algorithms depends on bounding the instantaneous regret for each action. The following result gives an upper bound on the instantaneous regret when using a contextual algorithm $\mathfrak{A}$.

**Lemma 4.** *Let $G$ be the goodness function defined in Eq. (2) and $\mathfrak{A}$ be an OFD compatible contextual bandit algorithm with $|f_t^{\mathfrak{A}}(x_{t,n}) - f(x_{t,n})| \le h(x_{t,n}, \mathcal{O}_t)$. Then, the instantaneous regret incurred by* **OFD-$\mathfrak{A}$** *for selecting agent $n_t$ for item $m_t$ is*

$$r_t = G\left(TU_{t,n_t^\star}\right) - G\left(TU_{t,n_t}\right) \le 2 w_{n_t} h(x_{t,n_t}, \mathcal{O}_t).$$

*Proof.* Recall the definition of the goodness function, i.e., $G(TU_{t,n_t}) = \sum_{n \in \mathcal{N}} w_n \, \Phi(TU_{t,n_t})$.

$$
\begin{aligned}
r_t(\mathfrak{A}) &= G\left(TU_{t,n_t^\star}\right) - G\left(TU_{t,n_t}\right) \\
&= \sum_{n \in \mathcal{N}} w_n \, \Phi\left(TU_{t,n_t^\star}\right) - \sum_{n \in \mathcal{N}} w_n \, \Phi\left(TU_{t,n_t}\right) \\
&= \sum_{n \in \mathcal{N}} w_n \, \Phi^\star\left(TU_{t,n}\right) + w_{n_t^\star} f(x_{t,n_t^\star}) - \sum_{n \in \mathcal{N}} w_n \, \Phi\left(TU_{t,n_t}\right) \\
&\le \sum_{n \in \mathcal{N}} w_n \, \Phi^\star\left(TU_{t,n}\right) + w_{n_t^\star}\left[f_t^{\mathfrak{A}}(x_{t,n_t^\star}) + h(x_{t,n_t^\star}, \mathcal{O}_t)\right] - \sum_{n \in \mathcal{N}} w_n \, \Phi\left(TU_{t,n_t}\right) \\
&\le \sum_{n \in \mathcal{N}} w_n \, \Phi\left(TU_{t,n} + \left[f_t^{\mathfrak{A}}(x_{t,n_t^\star}) + h(x_{t,n_t^\star}, \mathcal{O}_t)\right] \mathbb{1}(n = n_t^\star)\right) - \sum_{n \in \mathcal{N}} w_n \, \Phi\left(TU_{t,n_t}\right) \\
&\le \sum_{n \in \mathcal{N}} w_n \, \Phi\left(TU_{t,n} + \left[f_t^{\mathfrak{A}}(x_{t,n_t}) + h(x_{t,n_t}, \mathcal{O}_t)\right] \mathbb{1}(n = n_t)\right) - \sum_{n \in \mathcal{N}} w_n \, \Phi\left(TU_{t,n_t}\right) \\
&\le \sum_{n \in \mathcal{N}} w_n \, \Phi^{f_t}\left(TU_{t,n} + \left[f_t^{\mathfrak{A}}(x_{t,n_t}) + h(x_{t,n_t}, \mathcal{O}_t)\right] \mathbb{1}(n = n_t)\right) - \sum_{n \in \mathcal{N}} w_n \, \Phi\left(TU_{t,n_t}\right) \\
&= \sum_{n \in \mathcal{N}} w_n \, \Phi^{f_t}\left(TU_{t,n}\right) + w_{n_t}\left[f_t^{\mathfrak{A}}(x_{t,n_t}) + h(x_{t,n_t}, \mathcal{O}_t)\right] - \sum_{n \in \mathcal{N}} w_n \, \Phi\left(TU_{t,n_t}\right) \\
&= \sum_{n \in \mathcal{N}} \cancel{w_n \, \Phi^{f_t}(TU_{t,n})} + w_{n_t}\left[f_t^{\mathfrak{A}}(x_{t,n_t}) + h(x_{t,n_t}, \mathcal{O}_t)\right] \\
&\qquad - \sum_{n \in \mathcal{N}} \cancel{w_n \, \Phi(TU_{t,n_t})} - w_{n_t} f(x_{t,n_t}) \\
&= w_{n_t}\left[f_t^{\mathfrak{A}}(x_{t,n_t}) + h(x_{t,n_t}, \mathcal{O}_t)\right] - w_{n_t} f(x_{t,n_t}) \\
&= w_{n_t}\left[f_t^{\mathfrak{A}}(x_{t,n_t}) - f(x_{t,n_t}) + h(x_{t,n_t}, \mathcal{O}_t)\right] \\
&\le w_{n_t}\left[|f_t^{\mathfrak{A}}(x_{t,n_t}) - f(x_{t,n_t})| + h(x_{t,n_t}, \mathcal{O}_t)\right]
\end{aligned}
$$

$$\leq w_{n_t} \left[ h(x_{t,n_t}, \mathcal{O}_t) + h(x_{t,n_t}, \mathcal{O}_t) \right]$$
$$= 2 w_{n_t} h(x_{t,n_t}, \mathcal{O}_t).$$

The $\Phi^\star(\cdot)$ returns the utilities in the same order as $\Phi\left(\mathrm{TU}_{t,n_t^\star}\right)$, while $\Phi^{f_t}(\cdot)$ returns the utilities in the same order as $\Phi\left(\mathrm{TU}_{t,n_t}\right)$. The first inequality is from applying $|f_t^{\mathfrak{A}}(x_{t,n}) - f(x_{t,n})| \leq h(x_{t,n}, \mathcal{O}_t)$ with $x_{t,n} = x_{t,n_t^\star}$ and then using triangle inequality to get $f(x_{t,n_t^\star}) \leq f_t^{\mathfrak{A}}(x_{t,n_t^\star}) + h(x_{t,n_t^\star}, \mathcal{O}_t)$. The second inequality follows from Lemma 3 as an additional utility of $f_t^{\mathfrak{A}}(x_{t,n_t^\star}) + h(x_{t,n_t^\star}, \mathcal{O}_t)$ is added to the total collected utility of agent $n_t^\star$. The third inequality follows from the fact that $n_t$ maximizes the goodness function. The fourth inequality follows from Lemma 2. The last inequality is due to $|f_t^{\mathfrak{A}}(x_{t,n}) - f(x_{t,n})| \leq h(x_{t,n}, \mathcal{O}_t)$. This completes the proof, showing that $r_t(\mathfrak{A}) \leq 2 w_{n_t} h(x_{t,n_t}, \mathcal{O}_t)$. $\qquad\square$

### C.1. Proof of Theorem 1

**Theorem 1.** *Let $\delta \in (0,1)$, $\lambda > 0$, $\|\theta^\star\|_2 \leq S$, $\|x_{t,n}\|_2 \leq L \ \forall t \geq 1, n \in \mathcal{N}$, noise in utility be the $R$-sub-Gaussian, and the goodness function be the same as defined in Eq. (2) with $w_{\max} = \max_{n \in \mathcal{N}} w_n$. Then, with a probability of at least $1 - \delta$, the regret in $T > 0$ rounds is*

$$\mathfrak{R}_T(\textbf{OFD-UCB}) \leq 2\alpha_T w_{\max} \sqrt{2dT \log(\lambda + TL^2/d)},$$

*where $\alpha_T = R\sqrt{d \log\left(\frac{1 + (TL^2/\lambda)}{\delta}\right)} + \lambda^{\frac{1}{2}} S$.*

*Proof.* As $h(x_{t,n_t}, \mathcal{O}_t) = \left( R\sqrt{d \log\left(\frac{1+(tL^2/\lambda)}{\delta}\right)} + \lambda^{\frac{1}{2}} S \right) \|x_{t,n_t}\|_{M_t^{-1}}$ when the linear contextual bandit algorithm is used. Applying Theorem 2 (proved independently below) with this choice of $h$, we have

$$\mathfrak{R}_T(\textbf{OFD-UCB}) \leq 2 w_{\max} \sqrt{T} \sqrt{\sum_{t=1}^{T} [h(x_{t,n_t}, \mathcal{O}_t)]^2}$$

$$= 2 w_{\max} \sqrt{T} \sqrt{\sum_{t=1}^{T} \left[ \left( R\sqrt{d \log\left(\frac{1+(tL^2/\lambda)}{\delta}\right)} + \lambda^{\frac{1}{2}} S \right) \|x_{t,n_t}\|_{M_t^{-1}} \right]^2}$$

$$= 2 w_{\max} \sqrt{T} \sqrt{\sum_{t=1}^{T} \left( R\sqrt{d \log\left(\frac{1+(tL^2/\lambda)}{\delta}\right)} + \lambda^{\frac{1}{2}} S \right)^2 \left[ \|x_{t,n_t}\|_{M_t^{-1}} \right]^2}$$

$$\leq 2 w_{\max} \sqrt{T} \sqrt{\sum_{t=1}^{T} \left( R\sqrt{d \log\left(\frac{1+(TL^2/\lambda)}{\delta}\right)} + \lambda^{\frac{1}{2}} S \right)^2 \left[ \|x_{t,n_t}\|_{M_t^{-1}} \right]^2}$$

$$= 2 w_{\max} \sqrt{T} \left( R\sqrt{d \log\left(\frac{1+(TL^2/\lambda)}{\delta}\right)} + \lambda^{\frac{1}{2}} S \right) \sqrt{\sum_{t=1}^{T} \left[ \|x_{t,n_t}\|_{M_t^{-1}} \right]^2}$$

$$= 2\alpha_T w_{\max} \sqrt{T} \sqrt{\sum_{t=1}^{T} \left[ \|x_{t,n_t}\|_{M_t^{-1}} \right]^2}$$

$$\leq 2\alpha_T w_{\max} \sqrt{T} \sqrt{2 \log \frac{\det(M_T)}{\det(\lambda I_d)}}$$

$$\leq 2\alpha_T w_{\max} \sqrt{2dT \log(\lambda + TL^2/d)}$$

$$\implies \mathfrak{R}_T(\textbf{OFD-UCB}) \leq 2\alpha_T w_{\max} \sqrt{2dT \log(\lambda + TL^2/d)}.$$

The last two inequalities follow from Lemma 11 and Lemma 10 of (Abbasi-Yadkori et al., 2011), respectively. With this, the proof of Theorem 1 is complete. $\qquad\square$

### C.2. Proof of Theorem 2

**Theorem 2.** *Let $\mathfrak{A}$ be an OFD compatible contextual bandit algorithm with $|f_t^{\mathfrak{A}}(x_{t,n}) - f(x_{t,n})| \leq h(x_{t,n}, \mathcal{O}_t)$ and the goodness function be same as defined in Eq. (2) with $w_{\max} = \max_{n \in \mathcal{N}} w_n$. If the assumptions underlying $\mathfrak{A}$ holds, then, with a probability of at least $1 - \delta$, the regret of corresponding OFD algorithm **OFD-$\mathfrak{A}$** in $T$ rounds is*

$$\mathfrak{R}_T(\textbf{OFD-}\mathfrak{A}) \leq 2w_{\max}\sqrt{T}\sqrt{\sum_{t=1}^{T}[h(x_{t,n_t}, \mathcal{O}_t)]^2}.$$

*Proof.* Let $r_t(\mathfrak{A})$ denote the instantaneous regret for using contextual bandit algorithm $\mathfrak{A}$. After using Lemma 4, the regret with contextual bandit algorithm $\mathfrak{A}$ after $T$ rounds is given as follows:

$$\mathfrak{R}_T(\mathfrak{A}) = \sum_{t=1}^{T} r_t(\mathfrak{A}) \leq 2\sum_{t=1}^{T} w_{n_t} h(x_{t,n_t}, \mathcal{O}_t) \leq 2w_{\max}\sum_{t=1}^{T} h(x_{t,n_t}, \mathcal{O}_t)$$

$$\leq 2w_{\max}\sqrt{T\sum_{t=1}^{T}[h(x_{t,n_t}, \mathcal{O}_t)]^2}$$

$$\implies \mathfrak{R}_T(\mathfrak{A}) \leq 2w_{\max}\sqrt{T}\sqrt{\sum_{t=1}^{T}[h(x_{t,n_t}, \mathcal{O}_t)]^2}. \qquad \square$$

### C.3. Proof of Theorem 3

**Theorem 3.** *Let $\mathfrak{A}$ be an OFD compatible contextual bandit algorithm with $|f_t^{\mathfrak{A}}(x_{t,n}) - f(x_{t,n})| \leq h(x_{t,n}, \mathcal{O}_t)$ and the goodness function G is locally monotonically non-decreasing and locally Lipschitz continuous, with $c_{\max} = \max_{n \in \mathcal{N}} c_n$. If the assumptions used in $\mathfrak{A}$ holds, then, with a probability of at least $1 - \delta$, the regret of corresponding OFD algorithm **OFD-$\mathfrak{A}$** in $T$ rounds is*

$$\mathfrak{R}_T(\textbf{OFD-}\mathfrak{A}, \textbf{G}) \leq 2c_{\max}\sqrt{T}\sqrt{\sum_{t=1}^{T}[h(x_{t,n_t}, \mathcal{O}_t)]^2}.$$

*Proof.* First, we get an upper bound on instantaneous regret:

$$\begin{aligned}
r_t(\mathfrak{A}) &= \textbf{G}\left(\text{TU}_{t,n_t^\star}\right) - \textbf{G}\left(\text{TU}_{t,n_t}\right) \\
&= \textbf{G}\left(U_{t,n_1}, \ldots, U_{t,n_t^\star} + f(x_{t,n_t^\star}), \ldots, U_{t,n_N}\right) - \textbf{G}\left(\text{TU}_{t,n_t}\right) \\
&\leq \textbf{G}\left(U_{t,n_1}, \ldots, U_{t,n_t^\star} + f_t^{\mathfrak{A}}(x_{t,n_t^\star}) + h(x_{t,n_t^\star}, \mathcal{O}_t), \ldots, U_{t,n_N}\right) - \textbf{G}\left(\text{TU}_{t,n_t}\right) \\
&\leq \textbf{G}\left(U_{t,n_1}, \ldots, U_{t,n_t} + f_t^{\mathfrak{A}}(x_{t,n_t}) + h(x_{t,n_t}, \mathcal{O}_t), \ldots, U_{t,n_N}\right) - \textbf{G}\left(\text{TU}_{t,n_t}\right) \\
&= \textbf{G}\left(U_{t,n_1}, \ldots, U_{t,n_t} + f_t^{\mathfrak{A}}(x_{t,n_t}) + h(x_{t,n_t}, \mathcal{O}_t), \ldots, U_{t,n_N}\right) \\
&\qquad - \textbf{G}\left(U_{t,n_1}, \ldots, U_{t,n_t} + f(x_{t,n_t}), \ldots, U_{t,n_N}\right) \\
&\leq c_{n_t}|U_{t,n_t} + f_t^{\mathfrak{A}}(x_{t,n_t}) + h(x_{t,n_t}, \mathcal{O}_t) - (U_{t,n_t} + f(x_{t,n_t}))| \\
&= c_{n_t}\left[f_t^{\mathfrak{A}}(x_{t,n_t}) - f(x_{t,n_t}) + h(x_{t,n_t}, \mathcal{O}_t)\right] \\
&\leq c_{n_t}\left[|f_t^{\mathfrak{A}}(x_{t,n_t}) - f(x_{t,n_t})| + h(x_{t,n_t}, \mathcal{O}_t)\right] \\
&\leq c_{n_t}\left[h(x_{t,n_t}, \mathcal{O}_t) + h(x_{t,n_t}, \mathcal{O}_t)\right] = 2c_{n_t}h(x_{t,n_t}, \mathcal{O}_t).
\end{aligned}$$

The first inequality follows from the monotonicity property of goodness function after applying $|f_t^{\mathfrak{A}}(x_{t,n}) - f(x_{t,n})| \leq h(x_{t,n}, \mathcal{O}_t)$ with $x_{t,n} = x_{t,n_t^\star}$ and then using triangle inequality to get $f(x_{t,n_t^\star}) \leq f_t^{\mathfrak{A}}(x_{t,n_t^\star}) + h(x_{t,n_t^\star}, \mathcal{O}_t)$. The second inequality follows from the fact that $n_t$ maximizes the goodness function. The third inequality is due to the locally Lipschitz property of the goodness function. Let $c_{\max} = \max\{c_1, c_2, \ldots, c_N\}$. Then, after $T$ rounds, the regret with contextual bandit algorithm $\mathfrak{A}$ is given as follows:

$$\mathfrak{R}_T(\mathfrak{A}) = \sum_{t=1}^{T} r_t(\mathfrak{A}) \leq 2\sum_{t=1}^{T} c_{n_t} h(x_{t,n_t}, \mathcal{O}_t)$$

$$\leq 2c_{\max}\sqrt{T\sum_{t=1}^{T}[h(x_{t,n_t},\mathcal{O}_t)]^2}$$

$$\implies \mathfrak{R}_T(\mathfrak{A}) \leq 2c_{\max}\sqrt{T}\sqrt{\sum_{t=1}^{T}[h(x_{t,n_t},\mathcal{O}_t)]^2}. \qquad \square$$

*Table 1.* Examples of different $h(x_{t,n},\mathcal{O}_t)$ values for some contextual bandit algorithms (using notations from original papers).

| Contextual bandit algorithm | $h(x_{t,n},\mathcal{O}_t)$ |
|---|---|
| Lin-UCB (Chu et al., 2011) | $\left(R\sqrt{d\log\left(\frac{1+\frac{tL^2}{\lambda}}{\delta}\right)} + \lambda^{\frac{1}{2}}S\right)\|x_{t,n}\|_{M_t^{-1}}$ |
| GLM-UCB (Li et al., 2017) | $\sqrt{\frac{d}{2}\log(1+2t/d)+\log(1/\delta)}\frac{\|x_{t,n}\|_{M_t^{-1}}}{\kappa}$ |
| IGP-UCB (Chowdhury & Gopalan, 2017) | $\sqrt{2(\gamma_{t-1}+1+\log(1/\delta))}\sigma_{t-1}(x_{t,n}) + B\sigma_{t-1}(x)$ |

## D. Additional Experimental Results

**Regret vs. number of agents** $(N)$ **and dimension** $(d)$**.** As shown in Fig. 2g and Fig. 6c, the regret bound of our UCB- and TS- based algorithms increases as we increase the number of agents, i.e., $N = \{5, 10, 15, 20, 25\}$. We also observe the same trend when we increase the dimension of the item-agent feature vector from $d = \{10, 20, 30, 40, 50\}$ as shown in Fig. 2h and Fig. 6d. In all experiments, we also observe that the TS-based algorithm performs better than its UCB-based counterpart (as shown in Fig. 5 and Fig. 6 by comparing their regret). However, when we set the value of $\rho = 0.85$, we observe the same trend for $d$, but the trend reverses as the number of agents increases due to the goodness function (defined in Eq. (2)) decreasing with an increase in the number of agents when $\rho < 1$, as shown in Fig. 5. It happens because the value of the goodness function decreases with an increase in the number of agents when $\rho < 1$.

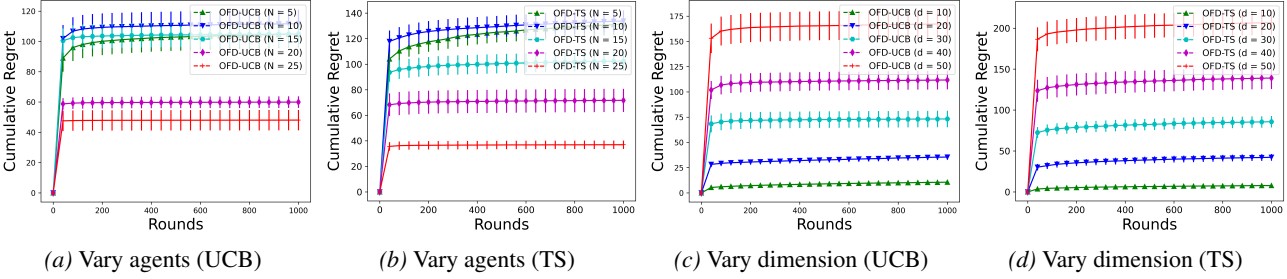

*(a)* Vary agents (UCB)       *(b)* Vary agents (TS)      *(c)* Vary dimension (UCB)      *(d)* Vary dimension (TS)

*Figure 5.* Cumulative regret of **OFD-UCB** and **OFD-TS** vs. different values of $N$ and $d$.

**Non-linear utility function.** For this experiment, we adapt problem instances with non-linear utility functions from those used for linear utility functions in Sec. 5. We apply a polynomial kernel of degree 2 to transform the item-agent feature vectors to introduce non-linearity. The constant terms (i.e., the 1's) resulting from this transformation are removed. As an example, a sample 4-d feature vector $x = (x_1, x_2, x_3, x_4)$ is transformed into a 14-d feature vector: $x' = (x_1, x_2, x_3, x_4, x_1x_2, x_1x_3, x_1x_4, x_2x_3, x_2x_4, x_3x_4, x_1x_2x_3, x_1x_2x_4, x_1x_3x_4, x_2x_3x_4)$. We also remove 1's, which appear in the transformed samples. As shown in Fig. 6, our algorithms (nOFD-UCB and nOFD-TS, prefixed with 'n') consistently outperform both baselines across various non-linear problem instances (where only the feature dimension $d = \{4, 6, 10, 20\}$ is varied, while all other parameters remain the same as the instances for linear utility functions). To better demonstrate the sub-linear regret behavior of our algorithms, we limit the y-axis in the plots. Additionally, we observe that the TS-based algorithm achieves lower regret than its UCB-based counterpart.

**Computational resources.** All experiments are run on a server with an AMD EPYC 7543 32-Core Processor, 256GB of RAM, and 8 GeForce RTX 3080 GPUs.

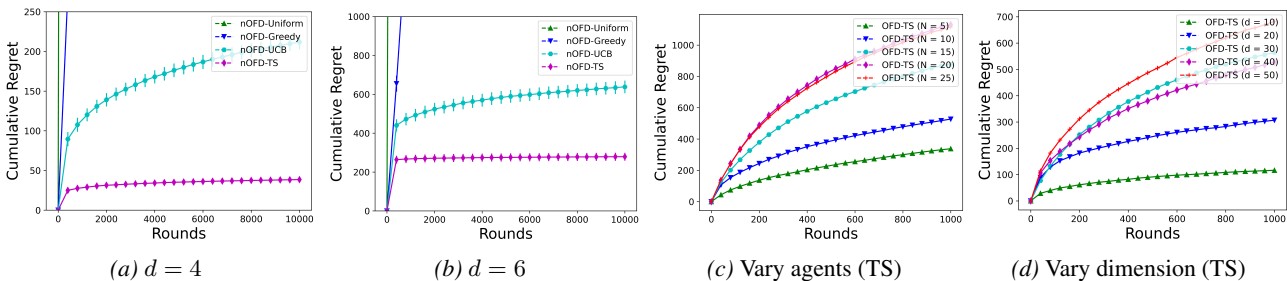

*Figure 6.* **(Fig. 6a–Fig. 6b):** Cumulative regret of **nOFD-UCB** and **nOFD-TS** on non-linear utility instances ($\rho = 0.85$) for varying dimension $d$. **(Fig. 6c–Fig. 6d):** Cumulative regret of **OFD-TS** vs. number of agents $N$ and dimension $d$ for $\rho = 1.0$.

## E. Auxiliary Observations

We first mention some of the observations that will be useful to prove the property of monotonicity and Lipschitz continuity for our various choices of the goodness function.

**Fact 1.** *Suppose $h(x)$ and $g(x)$ are Lipschitz continuous functions with non-negative bounded domains. Then the following statements are true:*

1. *$h(x) + g(x)$ is a Lipschitz continuous function.*

2. *$h(x)g(x)$ is a Lipschitz continuous function.*

*Proof.* Let $h(x)$ and $g(x)$ be the Lipschitz continuous functions in a bounded domain with Lipschitz constants $k_1$ and $k_2$, respectively.

**1.** We first prove that $h(x) + g(x)$ is a Lipschitz continuous function. Consider the following difference:

$$
\left| (h(x) + g(x)) - (h(y) + g(y)) \right| \leq \left| h(x) - h(y) \right| + \left| g(x) - g(y) \right|
$$
$$
\leq k_1 \left| x - y \right| + k_2 \left| x - y \right| \leq k \left| x - y \right|
$$

where $k = k_1 + k_2$. The first inequality follows from the triangle inequality, while the second inequality uses Lipschitz continuity.

**2.** We now prove that $h(x)g(x)$ is a Lipschitz continuous function. Consider the following difference:

$$
\left| h(x)g(x) - h(y)g(y) \right| = \left| h(x)g(x) - h(y)g(x) + h(y)g(x) - h(y)g(y) \right|
$$
$$
\leq \left| h(x)g(x) - h(y)g(x) \right| + \left| h(y)g(x) - h(y)g(y) \right|
$$
$$
\leq |g(x)| \left| h(x) - h(y) \right| + |h(y)| \left| g(x) - g(y) \right|
$$
$$
\leq Mk_1 \left| x - y \right| + Mk_2 \left| x - y \right| \leq M' \left| x - y \right|, \quad \text{where } M' = Mk.
$$

The last inequality is due to the bound of the continuous function, and appropriately, the Lipschitz constant is chosen. □

**Fact 2.** *Suppose $h_i(x)$ with $1 \leq i \leq n$ are Lipschitz continuous functions with a non-negative bounded domain. Then the following statements are true:*

1. *$\sum_{i=1}^{n} a_i h_i(x)$ is a Lipschitz continuous function, where all $a_i$'s are constant.*

**2.** $\prod_{i=1}^{n} h_i(x)$ *is a Lipschitz continuous function.*

*Proof.* **1.** We first prove that $\sum_{i=1}^{n} a_i h_i(x)$ is a Lipschitz continuous function. Consider the following difference:

$$
\left| \sum_{i=1}^{n} a_i h_i(x) - \sum_{i=1}^{n} a_i h_i(y) \right| = \left| \sum_{i=1}^{n} a_i \Big( h_i(x) - h_i(y) \Big) \right|
$$

$$
\leq \sum_{i=1}^{n} |a_i| \Big| h_i(x) - h_i(y) \Big|
$$

$$
\leq \sum_{i=1}^{n} k_i |a_i| \Big| x - y \Big| = L \Big| x - y \Big|.
$$

The first inequality is due to the triangle inequality, while the second inequality is due to the Lipschitz continuity of the function.

**2.** We now prove that $\prod_{i=1}^{n} h_i(x)$ is a Lipschitz continuous function. We will use induction for this proof. For base case: When $n = 2$, the statement is true due to Fact 1. For the induction hypothesis: Assume the statement is true for $n = m$. Now, we consider the inductive step (for $n = m + 1$) as follows:

$$
\left| \prod_{i=1}^{m+1} h_i(x) - \prod_{i=1}^{m+1} h_i(y) \right| = \left| \prod_{i=1}^{m+1} h_i(x) - h_{m+1}(y) \prod_{i=1}^{m} h_i(x) + h_{m+1}(y) \prod_{i=1}^{m} h_i(x) - \prod_{i=1}^{m+1} h_i(y) \right|
$$

$$
= \left| \Big( h_{m+1}(x) - h_{m+1}(y) \Big) \prod_{i=1}^{m} h_i(x) + h_{m+1}(y) \Big( \prod_{i=1}^{m} h_i(x) - \prod_{i=1}^{m} h_i(y) \Big) \right|
$$

$$
\leq \left| \Big( h_{m+1(x)} - h_{m+1}(y) \Big) \prod_{i=1}^{m} h_i(x) \right| + \left| h_{m+1}(y) \Big( \prod_{i=1}^{m} h_i(x) - \prod_{i=1}^{m} h_i(y) \Big) \right|
$$

$$
= \left| \prod_{i=1}^{m} h_i(x) \right| \left| \Big( h_{m+1}(x) - h_{m+1}(y) \Big) \right| + \left| h_{m+1}(y) \right| \left| \Big( \prod_{i=1}^{m} h_i(x) - \prod_{i=1}^{m} h_i(y) \Big) \right|
$$

$$
\leq \beta \Big| x - y \Big|.
$$

The first inequality follows from the triangle inequality. The last inequality follows from the function's boundedness and the induction hypothesis. $\qquad\square$

**Fact 3.** *The function* $\log(x)$ *is Lipschitz continuous when $x$ is in a non-negative domain, i.e., $x > 0$.*

*Proof.* Suppose $0 < x \leq l$. Now consider the following difference.

$$
\left| \log(x) - \log(y) \right| = \left| \log\left( \frac{y}{x} \right) \right| = \left| \log\left( 1 + \frac{y}{x} - 1 \right) \right| \leq \left| \frac{y}{x} - 1 \right| \leq \frac{1}{l} \Big| x - y \Big|.
$$

We use that $z > \log(1 + z)$ for $z > -1$. Therefore, it proves our claim. $\qquad\square$

**Fact 4.** *Suppose $h(x)$ is a continuous function, bounded and positive, then $\log(h(x))$ is a Lipschitz continuous function.*

*Proof.* Suppose $0 < h(x) \leq l$. Now consider the following difference.

$$
\left| \log\Big( h(x) \Big) - \log\Big( h(y) \Big) \right| = \left| \log\left( \frac{h(y)}{h(x)} \right) \right|
$$

$$
= \left| \log\left( 1 + \frac{h(y)}{h(x)} - 1 \right) \right|
$$

$$
\leq \left| \frac{h(y)}{h(x)} - 1 \right| \leq \frac{1}{l} \Big| h(x) - h(y) \Big|.
$$

Using Fact 3, the proof is complete. $\qquad\square$

## F. Goodness functions that are locally monotonically non-decreasing and Lipschitz continuous

In this section, we will briefly discuss the different choices of the goodness function for which a sub-linear regret upper bound can be theoretically guaranteed.

**Weighted Gini social-evaluation function.** This function is given as follows:

$$G\left(\mathrm{TU}_{t,n_t}\right) = \sum_{n \in \mathcal{N}} w_n \, \Phi\left(\mathrm{TU}_{t,n_t}\right).$$

Since all $w_n$'s are non-negative constants with $w_{\max} = \max_{n \in \mathcal{N}} w_n$ and $\Phi\left(\mathrm{TU}_{t,n_t}\right)$ are bounded, positive, continuous functions. Therefore, the weighted Gini social-evaluation function satisfies the locally Lipschitz condition. Also, if we change the $i$-th component while keeping the rest of the components of the weighted Gini social-evaluation function fixed, the locally monotonicity non-decreasing condition holds (also from Lemma 3). Therefore, Theorem 3 (with $c_{\max} = w_{\max}$) also holds for weighted Gini social-evaluation function.

**Targeted weights.** Let us define a fixed target weight vector in advance, with the learner's goal being to achieve these targeted weight distributions after each allocation. These target weights represent the desired fraction of the total cumulative utility each agent should receive. Let $\mathrm{SU}_{n_t} = f(x_{t,n_t}) + \sum_{n \in \mathcal{N}} U_{t,n}$ be the sum of total utility after allocating item $m_t$ to agent $n_t$. Suppose the target weighted fraction of the utility vector is $\mathbf{r}^*$, which is given. The learner's goal is to obtain the targeted weight vector at the end of the allocation process. The goodness function for this fairness constraint is defined as:

$$G\left(\mathrm{TU}_{t,n_t}\right) = \sum_{n \in \mathcal{N}} w_n \, \Phi(\mathrm{TU}_{t,n_t}^{\mathbf{P}}),$$

where $w_1 = 1$ and $w_i = 0$ for $i \geq 2$ [i.e., ESW]. The utility $U_{t,n}^{\mathbf{P}}$ in $\mathbf{U}_{t,n_t}^{\mathbf{P}}$ is defined as $U_{t,n}^{\mathbf{P}} = U_{t,n}/p_n$, where $p_n$ is the proportional ratio for the $n$-th agent. Let $\mathbf{r}^* = (r_1, \ldots, r_N)$ be the vector of targeted ratios of agents' cumulative utility to total utilities collected by all agents (i.e., system's total utility), where $\sum_{n=1}^{N} r_n = 1$. Then, $p_n = r_n / \min_i(r_i)$, e.g., if $N = 3$ and $\mathbf{r} = (0.2, 0.5, 0.3)$, then $\mathbf{p} = (0.2/0.2, 0.5/0.2, 0.3/0.2) = (1, 2.5, 1.5)$.

**Nash Social Welfare.** The Nash Social Welfare (NSW) is defined as the geometric mean of agents' utilities, i.e., $\left[\prod_{n \in \mathcal{N}} U_n\right]^{\frac{1}{|\mathcal{N}|}}$. Since the map $x \mapsto x^{1/|\mathcal{N}|}$ is monotone increasing for $x > 0$, maximizing NSW is equivalent to maximizing $\prod_{n \in \mathcal{N}} U_n$; hence we work with the product form for simplicity. First, observe that the utility function is $G(\cdot) : \mathbb{R}^n \mapsto \mathbb{R}^+$. Since the utility function is positive, NSW satisfies the monotonically non-decreasing function property. Also, we assume that the utility of the individual agent is positive and bounded, which implies that the NSW is locally Lipschitz continuous. As a result, the properties given in Definition 2 hold for NSW.

**Log Nash Social Welfare.** This function is commonly considered in the fair division literature (Cole & Gkatzelis, 2018; Talebi & Proutiere, 2018), and defined as follows:

$$G\left(\mathrm{TU}_{t,n_t}\right) = \log\left[\prod_{n \in \mathcal{N}} (U_{t,n} + f(x_{t,n})\mathbb{1}(n = n_t))\right] = \sum_{n \in \mathcal{N}} \log(U_{t,n} + f(x_{t,n})\mathbb{1}(n = n_t)).$$

We assume that the utility function is bounded and positive. It is easy to observe that using Fact 1- Fact 4, the log NSW is a Lipschitz continuous function. Also, log NSW satisfies the monotone property. By virtue of these observations, we can guarantee sub-linear regret, i.e., Theorem 3 holds.

## G. Frequently Asked Questions

**Question 1.** How this framework is different from the standard contextual bandit?

**Answer.** We emphasize that the central conceptual contribution of this work is the coupling of bandit exploration with social evaluation functions. In contrast to standard bandit settings, where the objective is to maximize a scalar reward, our algorithm seeks to optimize a global goodness function that evolves at each round as a function of the entire allocation history. Our analysis shows that the estimation errors inherent in contextual bandit learning can be systematically propagated through complex, non-linear social welfare functions, such as the Gini social evaluation function, while still guaranteeing sublinear regret, provided these functions satisfy appropriate Lipschitz continuity and monotonicity conditions.

**Question 2.** How would the regret bounds change if the number of copies were increased?

**Answer.** Our framework applies to settings in which only a single copy of each item is available. As the number of available copies increases, the regret associated with repeated allocations decreases due to the reduced uncertainty for those items, resulting in improved overall regret performance (see Fig. 2f). While we do not provide a theoretical characterization of this improvement, quantifying this effect constitutes an interesting direction for future work.

**Question 3.** Why the paper did not compare to the latest online fair division algorithms?

**Answer.** To the best of our knowledge, this is the first work in the contextual bandit setting to address online fair division with a large number of agents and limited item copies. Existing algorithms typically require a substantial number of copies per item to accurately estimate agent–item utilities and therefore are not directly applicable as baselines in our setting.

**Question 4.** What is the scalability of the proposed algorithm?

**Answer.** The scalability of the proposed algorithms in large-scale settings depends primarily on the choice of the underlying contextual bandit algorithm and the cost of evaluating the goodness function G for each agent. An appropriate bandit algorithm may be selected based on the application context, and the computation of G can be parallelized to mitigate computational overhead.

Our experimental objective is to empirically validate the theoretical results. To this end, we adopt standard experimental practices, including the use of synthetic benchmarks, and focus on representative settings studied in the paper.

**Question 5.** The approach critically depends on the choice of goodness function, but guidance on how to select or tune it in practice is limited.

**Answer.** We acknowledge that, in real-world deployments, the appropriate trade-off between fairness and efficiency is often determined by policy considerations rather than by a fixed mathematical constant. Accordingly, our framework is designed to be policy-agnostic. The goodness function serves as a plug-and-play interface: while we instantiate it using the weighted Gini social evaluation function, a standard economic measure of inequality, the framework readily accommodates alternative objectives. In particular, by adjusting the parameter $\rho$, practitioners can recover classical criteria ranging from utilitarianism to egalitarianism.

