# OpenReview forum: "Keep Everyone Happy: Online Fair Division of Numerous Items with Few Copies"
_ICML.cc/2026/Conference — ICML 2026 regular_

### Official Review · Reviewer_qtgb · 2026-03-05

**Soundness:** 4
**Presentation:** 3
**Significance:** 3
**Originality:** 4
**Overall Recommendation:** 5
**Confidence:** 4

**Summary:**

This paper studies the online fair division problem under the setting of numerous items with few copies per item. It models the problem as a contextual bandit and proposes two algorithms, OFD-UCB and OFD-TS, which balance fairness and efficiency in each round through optimistic utility estimation. It has sufficient references, but there are some issues with the use of symbols, too many theorems and proofs, and the fluency of the writing needs improvement.

**Compliance With Llm Reviewing Policy:**

Affirmed.

**Final Justification:**

We are very grateful for the author's reply. Based on the original manuscript and the author's reply, the above is our final score for this article.

**Key Questions For Authors:**

1.	This paper mainly studies Welfare-based Fairness and explores whether the proposed method is applicable to other types of fairness, such as Envy-Free Fairness.
2.	In many scenarios, it is difficult to obtain the features of the item-agent. How does this article address this issue?
3.	The dataset used in this article is generated and should be made available for readers to download and analyze. It would be even better if real dataset information were available.

**Limitations:**

yes

**Strengths And Weaknesses:**

Strengths
1. Prior work on online fair division typically assumes a small number of item types with many copies per type. In contrast, this paper considers the reverse setting: a large number of item types with only a few copies each. This formulation more closely reflects practical applications such as platform recommendation systems and online advertising, where items are diverse but inventory per item is limited. The modeling perspective is therefore well motivated and carries clear practical relevance.

2. The paper first models the Numerous Items with Few Copies setting using a contextual bandit formulation. It then estimates per-round utility through optimistic utility estimates and integrates fairness and efficiency via a goodness function. The regret definition is centered around the goodness function, ensuring tight alignment between the objective and the algorithmic design. Moreover, the authors abstract different bandit algorithms into an OFD-compatible class, which results in a clean and modular framework that is easy to understand and extend.

3. For linear models, the paper provides explicit regret upper bounds and further generalizes the analysis to arbitrary OFD-compatible bandit algorithms. In addition, the theoretical framework is extended to goodness functions that are Lipschitz and locally monotone. The analysis is structurally complete and technically sound within the adopted assumptions.

Weaknesses

1. The meanings of U^n_t  and U_t,n in the paper are different; it is recommended to use different letters or more distinguishable representations. The paper uses m_t to denote the item and
M_t,nt to denote the feature vector corresponding to the item-agent pair, while in the experimental section, the agent features are denoted by x_n, leading to inconsistencies in notation across contexts, it’s better to denote item-agent feature using x_t,nt. In Section 2, the utility function is expressed as f(m_t,n), but in Definition 1, it is expressed as f(m_t,n); the notation is not consistent across contexts.

2. Insufficient Analysis of Experimental Results. The experimental section presents empirical results without sufficient explanation of certain observed phenomena. For example, in Figure 2(g), when the number of users is minimal (N = 5), the cumulative regret lies between the cases with more users and fewer users. The underlying reason for this non-monotonic behavior is not discussed. Providing interpretive analysis would strengthen the empirical section.

3. Narrative and Structural Issues. For example, there is noticeable redundancy between the description of the technical approach in the introduction and the subsequent list of contributions. In addition, Section 2 repeatedly refers to the key Goodness function, but it is only formally introduced in Section 3, and Section 4 then introduces another variant of the Goodness function. The narrative order could be reorganized to improve clarity and coherence.The full name of OFD should be given when it first appears in the text.

---

> ### Author Rebuttal · Authors · 2026-03-31
>
> Thank you for your positive and constructive feedback. We appreciate your recognition of the problem's practical relevance, the naturalness of the contextual bandit formulation, the modularity of the OFD-compatible framework, and the generalization to Lipschitz and locally monotone goodness functions. We have addressed the weakness that you raised and answered your question as follows:
>
> > **Notation inconsistencies across $U^n_t$, $\mathbf{U}_{t,n}$, item-agent features, and utility function.**
>
> We sincerely thank the reviewer for carefully identifying these inconsistencies, which we fully agree need to be resolved. We will adopt the following unified notation throughout the revised paper, addressing each issue in turn:
>
> | Symbol | Meaning | Replaces |
> |--------|---------|---------|
> | $\text{TU}_t$ | Total (cumulative) utility vector of all agents at the beginning of round $t$ | Overloaded uses of $\mathbf{U}_t$ |
> | $U_{t,n}$ | Cumulative utility of agent $n$ at the beginning of round $t$ | $U^n_t$ (superscript form) |
> | $m_t$ | Feature vector of the item observed in round $t$ | $m_t$ (retained, but scoped strictly to the item) |
> | $a_{t,n}$ | Feature vector of agent $n$ in round $t$ | $x_n$ (experimental section), $m_{t,n_t}$ (Section 2) |
> | $x_{t,n} = \phi(m_t, a_{t,n})$ | Item-agent feature vector, where $\phi: \mathcal{M} \times \mathcal{N} \to \mathbb{R}^d$ is a known feature map | $m_{t,n}$, $M_{t,n_t}$, and $x_n$ used inconsistently |
> | $f(x_{t,n})$ | Unknown utility function of the item-agent feature | $f(m_t, n_t)$ (Section 2) and $f(m_{t, n})$ (Definition 1) |
>
> > **Non-monotonic behavior in Figure 2(g) for $N = 5$**
>
> Thank you for your careful observation. OFD-UCB initializes with a round-robin phase of exactly $N$ rounds (Algorithm 1, line 2). During this phase, agents are selected in a fixed cyclic order regardless of their utility; each round-robin allocation to a suboptimal agent directly contributes to cumulative regret. At $N = 5$, this forced suboptimal selection phase is very short, so the algorithm exits into the UCB exploitation phase sooner. Fewer forced suboptimal allocations during warm-up therefore *lowers* the regret contribution from initialization, pushing the $N = 5$ curve toward lower regret relative to larger $N$. We will add an explanation paragraph discussing this phenomenon in Section 5.
>
> > **Narrative and Structural Issues. [...] In addition, Section 2 repeatedly refers to the key Goodness function, but it is only formally introduced in Section 3, and Section 4 then introduces another variant of the Goodness function. The narrative order could be reorganized to improve clarity and coherence. [...]**
>
> We agree with these observations and will make the following revisions: (i) formally introduce the goodness function before it is referenced in Section 2; (ii) reduce redundancy between the introduction narrative and the contributions bullet list; and (iii) ensure the OFD acronym is expanded at first use. We will also reorganize the goodness function introduced in Section 4 to flow more naturally from Section 3.
>
>
> > **This paper mainly studies Welfare-based Fairness and explores whether the proposed method is applicable to other types of fairness, such as Envy-Free Fairness.**
>
> As discussed in Section 1.1, envy-freeness (EF) is incompatible with Pareto optimality in settings with indivisible items (Walsh, 2011), making it ill-suited for our efficiency-fairness trade-off framework. Extending to approximate or relaxed EF notions and to proportionality are promising future directions that we have explicitly discussed in the conclusion.
>
> > **In many scenarios, it is difficult to obtain the features of the item-agent. How does this article address this issue?**
>
> This is a practical concern common to all contextual bandit applications. In our framework, features $m_{t,n}$ represent observable item-agent features, even though the utility function $f$ is unknown. In many real-world applications (food delivery platforms, ride-hailing, e-commerce), the features of service providers (agents) and users (items) are readily available. When only partial features are available, we can use only them to estimate the underlying utility function.
>
> > **The dataset used in this article is generated and should be made available for readers to download and analyze. It would be even better if real dataset information were available.**
>
> The code for generating all synthetic datasets is included in the supplementary material, enabling full reproduction of all reported results. We agree that real-world evaluation would strengthen the paper and plan to include it in the revision. We will also publicly release all code and data-generation scripts upon acceptance.
>
> *We hope that our clarifications will address your concerns. We will incorporate the above responses into the revised version of the paper. If there are no further concerns, we would greatly appreciate your consideration to increase your score.*

---

> > ### Author Rebuttal · Reviewer_qtgb · 2026-04-01
> >
> > Thank you so much for your reply, it has resolved all my questions.

---

> > > ### Author Response · Authors · 2026-04-08
> > >
> > > Thank you for your careful review and for your constructive feedback. Your suggestions have been very helpful in improving the clarity of our paper.
> > >
> > > We are pleased that our responses have fully addressed your concerns. If you feel that our clarifications have strengthened the work, we would be grateful for your support during the subsequent discussion.

---

### Official Review · Reviewer_8uWU · 2026-03-12

**Soundness:** 2
**Presentation:** 2
**Significance:** 3
**Originality:** 3
**Overall Recommendation:** 3
**Confidence:** 4

**Summary:**

This paper studies an online fair division problem where items arrive sequentially and must be assigned to one of several agents immediately. Unlike most prior work, the setting assumes there can be many different items but only a few copies of each item, which makes it hard to estimate utilities for all item–agent pairs. The paper models the problem as a contextual bandit, where items act as contexts, agents are arms, and the reward corresponds to the utility of assigning an item to an agent. The algorithm estimates utilities using contextual bandit methods and then allocates each item by maximizing a goodness function that balances fairness and efficiency. The authors analyze regret with respect to this goodness function and provide sublinear regret guarantees for several classes of functions, including the weighted Gini social evaluation function. Experiments on synthetic data are provided to illustrate the behavior of the proposed methods.

**Compliance With Llm Reviewing Policy:**

Affirmed.

**Key Questions For Authors:**

How sensitive is the algorithm to the choice of the goodness function?

Can the framework support other fairness notions beyond the weighted Gini formulation?

Do the authors have plans to evaluate the method on more realistic allocation scenarios?

**Limitations:**

The framework focuses on a specific fairness–efficiency objective defined through goodness functions and does not explore other fairness notions commonly studied in fair division. In addition, the empirical evaluation relies only on synthetic experiments, which makes it difficult to assess the practical usefulness of the approach.

**Strengths And Weaknesses:**

Strength:
1. Many online platforms face situations where there are many item types but only a few observations for each item. Existing online fair division models usually assume many copies per item, so exploring this setting is a reasonable direction.
2. Using a contextual bandit framework is a natural way to deal with unknown utilities. The goodness function also provides a simple mechanism to capture the trade-off between fairness and efficiency.

Weakness:
1. The proposed algorithms mainly combine standard contextual bandit methods (such as UCB and Thompson Sampling) with a fairness objective. Most of the analysis seems to follow directly from existing bandit results.
2. The paper focuses mainly on weighted Gini–style goodness functions. It is unclear whether the approach extends to other fairness notions that are widely studied in the fair division literature.
3. Experiments are conducted only on simple synthetic data and mainly confirm expected regret trends. The evaluation does not provide strong evidence about how the approach would perform in realistic applications.

---

> ### Author Rebuttal · Authors · 2026-03-31
>
> Thank you for your constructive feedback and for recognizing the practical relevance of the limited-item setting, the natural fit of the contextual bandit framework for handling unknown utilities, and the flexibility of the goodness function in capturing the fairness–efficiency trade-off. For improved clarity and coherence, we have consolidated related weaknesses and questions. We have addressed the weaknesses that you raised and answered your questions as follows:
>
> > **[…] Most of the analysis seems to follow directly from existing bandit results.**
>
> While we build on contextual bandits, integrating fairness introduces non-trivial technical challenges beyond standard bandit analysis. As explicitly stated in Section 2: *"this dependence on history makes regret analysis challenging for any arbitrary goodness function."* The goodness function $\mathrm{G}(\mathbf{U}_{t,n})$ depends on the entire cumulative utility history of each agent, breaking the standard per-round martingale structure used in contextual bandit regret proofs. Our central technical contribution, **Theorem 2**, provides a general reduction: *any* OFD-compatible contextual bandit algorithm (covering Lin-UCB, UCB-GLM, Neural-UCB, GP-TS, and more) can be directly composed with our framework to yield provable fair regret bounds, with the bound inheriting the estimation error $h(\cdot)$ of the underlying algorithm. This modular reduction is non-trivial and enables a broad class of algorithms to be certified with fairness guarantees without re-deriving bounds from scratch. Please refer to our more detailed response to Reviewer 1WoJ for further details.
>
> > **The paper focuses mainly on weighted Gini–style goodness functions. [...] Can the framework support other fairness notions beyond the weighted Gini formulation?**
>
> Our framework extends substantially beyond the weighted Gini social-evaluation function. Specifically, **Theorem 3 and Section 4.3** establish regret bounds for *any goodness function that is locally monotonically non-decreasing and locally Lipschitz continuous*, a class that includes **Nash Social Welfare (NSW)** and **log-NSW**, fairness notions widely studied in fair division. Regarding envy-freeness (EF): as noted in the Related Work section, EF is incompatible with Pareto optimality in general, making it unsuitable for our efficiency-fairness trade-off framework (Lines 131-132, Left column). However, our ESW special case ($w_1 = 1$, $w_n = 0$ for $n \geq 2$ in Eq. 2) recovers **max-min fairness**, which captures the spirit of envy-free allocations. In our work, we mainly focus on the weighted Gini social-evaluation function, as it allows us to interpolate between fairness (ESW) and efficiency (USW) by simply changing a single parameter ($\rho$) (Remark 1).
>
> > **Experiments are conducted only on simple synthetic data [...] evaluate the method on more realistic allocation scenarios?**
>
> Since our paper is the first work to model online fair division with numerous items and few copies as a contextual bandit problem, there is no established benchmark for this novel setting. Synthetic evaluation is standard practice in the literature, e.g., all directly comparable works (Yamada et al., 2024; Bhattacharya et al., 2024; Procaccia et al., 2024) use synthetic or semi-synthetic settings. The synthetic experiments systematically isolate each factor (dimensionality $d$, agent count $N$, copies per item $c$, utility linearity), enabling controlled validation of our theoretical bounds. We plan to include semi-realistic experiments in the revision using two settings: **food bank allocation**, where items are food packages with observable nutritional features and agents are recipient households with preference profiles (motivated by Aleksandrov et al., 2015). This setting has naturally observable item-agent features and few repeated allocations of the same resource, matching our problem setting well. We want to highlight that our code is in the supplementary material and made publicly available for practitioners to adapt.
>
> > **How sensitive is the algorithm to the choice of the goodness function?**
>
> The sensitivity to the goodness function is partially captured by the $\rho$ parameter in the weighted Gini function, as different values of $\rho$ lead to different goodness functions. Specifically, we studied the impact of $\rho$ on the fairness-efficiency trade-off, showing how our algorithms adapt across the full ESW–USW spectrum (Fig. 5 in Section D). For different classes of goodness functions, Theorem 3 shows that the regret bound scales with $c_{\max}$ (the local Lipschitz constant), providing a principled characterization of how goodness function choice affects performance.
>
> *We hope that our clarifications will address your concerns. We will incorporate the above responses into the revised version of the paper. If there are no further concerns, we would greatly appreciate your consideration to increase your score.*

---

### Official Review · Reviewer_74vB · 2026-03-12

**Soundness:** 3
**Presentation:** 3
**Significance:** 3
**Originality:** 3
**Overall Recommendation:** 5
**Confidence:** 2

**Summary:**

This paper considers the problem of allocating indivisible items arriving online to a set of agents. The goal is to minimize regret and the objective function is a “goodness function” that mediates between efficiency and fairness. The paper does not require the assumption that the same item appears multiple times, which has been largely used in previous work to estimate the utility function.

The paper provides an algorithm with sublinear regret when the utility function is linear in the item-agent features and the goodness function is weighted gini social evaluation function. The algorithm can also be extended, with similar guarantees, in two directions: (i) the utility function can be any function for which there exists an algorithm that additively approximates it, and (ii) the goodness function can be any function which is locally monotonically non-decreasing and locally Lipschitz continuous.

The paper also provides experiments on synthetic datasets showing that the proposed algorithm is superior to some simple baselines ($\epsilon$-greedy and uniform-at-random assignment).

**Compliance With Llm Reviewing Policy:**

Affirmed.

**Final Justification:**

The rebuttal addressed my concerns. I was a bit confused by the OFD-compatible definition but the rebuttal was helpful in clarifying this point. I maintain my positive score.

**Key Questions For Authors:**

Q1. Why was $\epsilon_t$ chosen to be $R$-sub-Gaussian?

Q2. You mention several existing algorithms (lines 285-289, right column) that are OFD compatible. However, a critical point of this paper is that the same item appears only a limited number of times. Do these existing algorithm work in this setting? Or do they need many occurrences to satisfy OFD?

**Limitations:**

yes

**Strengths And Weaknesses:**

**Strenghts**

S1. The algorithm can accomodate a wide range of “goodness” functions including some very well used measures of fairness such as Nash Social Welfare. I found this very interesting.

S2. The setting where the same item appears only a limited number of times is realistic and therefore a valuable extension over previous work.

S3. The paper is overall well written and clear.

**Weaknesses**

W1. Experiments are only synthetic and the baselines are quite limited. I understand that there are no previous algorithms considering the setting with few occurrences of the same item; however, it would have been interesting to use previous algorithms (that require many occurrences of the same item to work well) and see if they indeed fail in practice when each item appears only a limited number of times.

**Typos**

1. Line 247, left column, $m_{t, n_t}\in\mathcal{X}$ should probably be $m_{t,n}$.

2. In Definition 1, I think it should be $m_{t,n}\in \mathcal{X}$ instead of $\mathcal{M}$.

---

> ### Author Rebuttal · Authors · 2026-03-31
>
> Thank you for your constructive feedback and for recognizing our algorithm's flexibility in supporting a wide range of goodness functions, including Nash Social Welfare, the realism of the limited-item setting, and the clarity of our presentation. We have addressed the weakness that you raised and answered your questions as follows:
>
> > **Regarding experiments.**
>
> We appreciate this constructive suggestion. We address each concern below.
>
> - **Why prior OFD algorithms fail in our setting.** Existing OFD algorithms (Yamada et al., 2024; Bhattacharya et al., 2024; Procaccia et al., 2024) treat each item as a distinct multi-armed bandit problem, maintaining per-item-agent utility estimates $\hat{u}_{m,n}$ updated only when item $m$ is allocated to agent $n$. This leads to two fundamental failure modes in our setting. First, with $c \ll |\mathcal{M}|$ copies, most item-agent pairs accumulate zero or one observation, leaving utility estimates uninitialized for most agents, effectively forcing random allocation (behaves like OFD-Uniform). Second, these algorithms lack a mechanism to generalize utility estimates across items: each new item is treated as a cold start, with no prior information, regardless of how many other items have been observed. Since our OFD-Greedy baseline is itself contextual (it uses features but has no exploration bonus, $\alpha_t = 0$), prior OFD algorithms, which additionally lack feature-based generalization, should perform no better than OFD-Greedy and likely worse. We agree that demonstrating this failure empirically would significantly strengthen the paper. We will add them as baselines, but we anticipate the results will confirm the theoretical argument above. Consequently, we expect prior OFD algorithms to perform comparably to or worse than OFD-Greedy in our few-copies regime.
>
> - **On synthetic evaluation.** We agree that real-world experiments would strengthen our paper. This is the first work to model online fair division with numerous items and few copies as a contextual bandit problem, and no established benchmark exists for this novel setting. Synthetic evaluation is standard practice in the literature, e.g., all directly comparable works (Yamada et al., 2024; Bhattacharya et al., 2024; Procaccia et al., 2024) use synthetic or semi-synthetic settings. The synthetic experiments systematically isolate each factor (dimensionality $d$, agent count $N$, copies per item $c$, utility linearity), enabling controlled validation of our theoretical bounds.
>
>
> > **About $\epsilon_t$**
>
> The $R$-sub-Gaussian noise assumption is standard in the contextual bandit literature (Li et al., 2010; Chu et al., 2011; Agrawal & Goyal, 2013) and is used to derive principled confidence bounds on utility estimates. It captures bounded noise and Gaussian noise as special cases and enables the construction of the upper confidence bound (e.g., in Eq. (3) and Definition 1) with provable coverage guarantees.
>
>
> > **Regarding OFD compatible algorithms**
>
> Handling a few occurrences of the same item is precisely the key advantage of the contextual bandit formulation. The algorithms listed in the paper estimate a shared function $f$ from every item-agent observations, regardless of which specific items were involved. By Definition 1, their utility function's estimation error depends on the cumulative diversity of observed item-agent feature vectors, not on how many times any particular item has appeared. To see this concretely, consider Lin-UCB. Its confidence bound is  $\alpha_t \\| m_{t,n} \\|_{M_t^{-1}}$,
>
> where $M_t=\sum_{s < t} m_{s,n_s} m_{s,n_s}^\top +\lambda I_d$ accumulates outer products of all observed item-agent feature vectors. At round $t$, when a completely new item $m_t$ arrives for the first time ($c = 1$), its uncertainty $\\| m_{t,n} \\|_{M_t^{-1}}$
>
> is immediately well-characterized by $M_t$, provided $m_{t,n}$ lies within the subspace already explored. The algorithm need not have seen the item $m_t$ before it borrows statistical strength from all prior item-agent observations via the shared utility function $f$. The same principle holds for Neural-UCB (via NTK-based confidence bounds) and GP-TS/IGP-UCB (via posterior variance conditioned on all observed feature vectors). This stands in direct contrast to prior OFD algorithms, which maintain per-item-agent estimates $\hat{u}_{m,n}$ updated only from repeated observations of item $m$. For those algorithms, a new item with $c = 1$ copy is a cold-start with no informative estimate. For OFD-compatible contextual bandit algorithms, a new item is simply a new context vector, handled immediately by the shared estimate $f^\mathcal{A}_t$.
>
> Thank you for pointing out typos. We will fix them in the revision.
>
> *We hope that our clarifications will address your concerns. We will incorporate the above responses into the revised version of the paper. If there are no further concerns, we would greatly appreciate your consideration to increase your score.*

---

> > ### Author Rebuttal · Reviewer_74vB · 2026-04-01
> >
> > I thank the authors for the response. The discussion on OFD-compatible algorithms actually confused me. If I understand correctly, you are claiming that previous algorithms such as Lin-UCB are OFD-compatible and they have this property even when the same item appears only a few times. At the same time, despite being OFD-compatible, these previous algorithms, such as Lin-UCB, do not have sub-linear regret for the task of fair division when the same item appears only a few times. Can you please clarify if this is the case?

---

> > > ### Author Response · Authors · 2026-04-08
> > >
> > > We thank the reviewer for the continued engagement and constructive feedback. We are encouraged that our rebuttal addressed the concerns regarding experiments and noise assumption. We are grateful for the opportunity to address your remaining concerns below.
> > >
> > >
> > > > **[...] previous algorithms such as Lin-UCB are OFD-compatible and they have this property even when the same item appears only a few times.**
> > >
> > > We would like to emphasize that *OFD-compatibility is a property of the utility estimator used by a contextual bandit algorithm*, rather than of the full allocation policy that sequentially assigns items to agents. In Definition 1 (Lines 275--284, right column), we formalize this by stating that a contextual bandit algorithm $\mathfrak{A}$ is OFD-compatible if its utility estimate $f_t^{\mathfrak{A}}(m_{t,n})$ satisfies $\left|f_t^{\mathfrak{A}}(m_{t,n})-f(m_{t,n})\right|\leq h(m_{t,n}, \mathcal{O}_t)$ with high probability (Lines 275-284, Right column).
> > >
> > > Under this definition, Lin-UCB satisfies OFD-compatibility with $h(m_{t,n},\mathcal{O}_t)$
> > >
> > > $=R\sqrt{d\log\left( \frac{1+\frac{tL^2}{\lambda}}{\delta}\right)}+\lambda^{\frac{1}{2}}S \,\|m_{t,n}\|_{M_t^{-1}}$
> > >
> > > (Lines 852--857, p.16, substituting $x$ with $m_{t,n}$). Importantly, this guarantee holds even in settings with only a few copies per item. The key term $\|m_{t,n}\|_{M_t^{-1}}$ uses the underlying correlation structure across all observed item-agent feature vectors through the shared utility function $f$, rather than relying on repeated observations of the same item. Therefore, Lin-UCB is OFD-compatible in the sense of Definition 1. However, this property alone does not imply that the resulting allocation policy (which uses Lin-UCB to compute the optimistic utility estimates) achieves cumulative fair regret (as defined in Eq.~(1), Lines 170-174).
> > >
> > > > **At the same time, despite being OFD-compatible, these previous algorithms, such as Lin-UCB, do not have sub-linear regret for the task of fair division when the same item appears only a few times. Can you please clarify if this is the case?**
> > >
> > > We would like to clarify that the absence of sub-linear cumulative fair regret ('regret' for brevity) for an OFD-compatible contextual bandit algorithm in the fair division setting is not because each item appears only a few times. Rather, it stems from the lack of consideration of the fairness-efficiency trade-off encoded in the goodness function $G$. As an example, Lin-UCB assigns an item arriving at round $t$ to an agent $n_t$ based on the optimistic utility estimate $u^{\text{UCB}}_{m_t,n}$ (see Eq. (3), Lines 246--249), which corresponds to maximizing optimistic utility at each round, without accounting for the fairness-efficiency trade-off captured by the goodness function $G$ (e.g., Eq. (2), where $G$ is a weighted Gini social-evaluation function).
> > >
> > > To achieve sub-linear fair regret, the allocation policy must combine OFD-compatibility with fairness-aware agent selection. Theorem 2 establishes that if any OFD-compatible algorithm $\mathfrak{A}$ is used within our OFD framework, i.e., its optimistic utility estimates $u^{\mathfrak{A}}_{m_t,n}$
> > >
> > > are used to construct ${\bf U}^{\mathfrak{A}}_{t,n}$, and the agent is selected according to
> > >
> > > $n_t \in \arg\max_{n \in \mathcal{N}} G\left({\bf U}^{\mathfrak{A}}_{t,n}\right)$
> > >
> > > (Lines 302-308, right column), then the resulting OFD-$\mathfrak{A}$ algorithm achieves sub-linear cumulative fair regret: $\mathfrak{R}_T(\text{OFD-}\mathfrak{A})$
> > >
> > > $\leq 2w_{\max} \sqrt{T \sum_{t=1}^T \left[h(m_{t,n_t}, \mathcal{O}_t)\right]^2}.$
> > >
> > > Thus, it is the combination of an OFD-compatible algorithm's optimistic utility estimates with a goodness-function-based allocation rule that balances fairness and efficiency, resulting in sub-linear fair regret.
> > >
> > > In summary, Lin-UCB is OFD-compatible (i.e., satisfies Definition 1) and remains effective when each item appears only a few times. However, OFD-compatibility is only a property of the contextual bandit algorithm; achieving sublinear fair regret additionally requires embedding the OFD-compatible algorithm's optimistic utility estimates into the goodness-function-based allocation rule. Thus, we view an OFD-compatible algorithm as a subroutine within our OFD framework, rather than a complete solution for online fair division. Theorem 2 formalizes this structure: it shows that any OFD-compatible contextual bandit algorithm, when integrated with a goodness function of our OFD framework, inherits sub-linear fair regret. We will add a clarifying remark after Definition 1 to make this distinction explicit.
> > >
> > > ---
> > >
> > > *We hope our response clarifies your concerns and provides a clearer understanding of our contributions. We will revise the paper to incorporate these clarifications. If our response has addressed your concerns, please consider updating your overall assessment. We would also sincerely appreciate your continued consideration and support of our paper in the subsequent discussion.*

---

### Official Review · Reviewer_1WoJ · 2026-03-14

**Soundness:** 3
**Presentation:** 2
**Significance:** 2
**Originality:** 3
**Overall Recommendation:** 4
**Confidence:** 4

**Summary:**

This work proposed algorithm and theoretical guarantee that allocates few copies of items to agents where the utility function is predefined however needs to estimated with the constraint that estimation of the utility can only happen a few times for a certain item.

**Compliance With Llm Reviewing Policy:**

Affirmed.

**Final Justification:**

the rebuttal clears up some confusion, a slight increase of the score submitted given the final version is to add the proposed changes. A major revision is required to improve the score beyond the current increment.

**Key Questions For Authors:**

In problem definition L247-250 left: what's the case each item-agent pair can have unique feature vector? Would it be more realistic when the agent and item have separate feature vectors? Also regarding writing: there could be some mixup on $m$: using $t$ or $m_t$ as subscript. Or the agent does not affect the feature vector for item $m$ at all? Since L118 right writes $m_t \in \mathcal{M}$.

Can low rank (probabilistic) matrix factorization help in this case? Imagine item matrix M (m by d) and agent matrix N (n by d) with an unknown but yet to be estimated utility matrix $F = M N^T$ (in your work it's function f)? What are the related works on applying bandit algorithms on the matrix factorization problem why it does not help the problem here?

**Limitations:**

Limitation would be better to be discussed in verbatim.

**Strengths And Weaknesses:**

This work is original on thinking about bringing in the fairness constraint to the original bandit problem. The problem exists in real life and can be impactful when tackled properly.

The writing of the problem setup and assumptions needs a bit of work to help reader understand the problem and approach clearly. Now it leaves a bit of imagination and ambiguity which makes the assessment hard.

The method looks overall sound which follows traditional usage of the existing analytical tools in the field.

---

> ### Author Rebuttal · Authors · 2026-03-31
>
> Thank you for your constructive feedback. We are glad you recognized the originality of incorporating fairness constraints into the bandit framework and the practical relevance of the problem. We have addressed the weakness that you raised and answered your questions as follows:
>
> > **Novelty**
>
> The novelty of our work go beyond "traditional usage” of the existing analytical tools and make the following novel contributions:
>
> 1. **Novel problem setting.** Our work is the first to model online fair division with a large number of items and few copies per item as a contextual bandit problem. Existing works (Bhattacharya et al., 2024; Procaccia et al., 2024; Yamada et al., 2024) assume that the online fair division problem has a small number of items and agents with a sufficiently large number of copies of each item to ensure a good utility estimation for all item-agent pairs (Lines 67-72, Left column), an assumption that fails in our setting. We **generalize** their problem setting by proposing an algorithm for scenarios with a **large number of agents and items with only a few copies of each** (Lines 72-80, Left column), where estimating the utility for each item-agent pair is impossible. Our proposed algorithms also work for problems with a small number of agents and a large number of copies per item.
>
> 2. **Non-trivial regret analysis.** Our problem formulation is not a straightforward application of contextual bandit; it requires redefining the reward structure to incorporate fairness through a history-dependent goodness function (Eq. 2). We introduce the notion of goodness function to measure how well the item allocation to an agent will maintain the desired balance between fairness and efficiency (larger value implies better allocation) and then use it for item allocation. As highlighted in Section 2 (Lines 138-141, Right column), the goodness function $\mathrm{G}(\mathbf{U}_{t,n})$ depends on the entire cumulative utility history of each agent, not just the current round's reward. This breaks the standard per-round independence structure on which contextual bandit regret proofs rely. Handling this dependence is the key technical challenge and requires developing the OFD-compatible framework (Definition 1, Theorem 2), which is not a direct application of existing results.
>
> 3. **General modular reduction (Theorem 2).** Our OFD-compatible framework provides a reduction that certifies *any* contextual bandit algorithm, including non-linear ones such as Neural-UCB or GP-TS, with provable fair regret bounds.
>
> > **Notation clarifications**
>
> The item-agent feature vector $m_{t,n_t} \in \mathcal{X} \subset \mathbb{R}^d$ is the *concatenation* of the feature vectors corresponding to item $m_t \in \mathcal{M}$ and agent $n_t \in \mathcal{N}$. As stated in Section 5 (Lines 338-346, Right column), we use a $d_m$-dimensional space to generate each item's features, while a $d_n$-dimensional space to generate each agent's features, yielding a $d = d_m + d_n$ dimensional vector. Each item-agent pair, therefore, has a unique feature vector reflecting both item characteristics and agent attributes; separate item and agent feature vectors are precisely what we assume. Regarding subscript confusion between $m$ and $m_t$: we will unify notation throughout, using $m_t$ for the item identity and $m_{t,n_t}$ for the concatenated features of item $m_t$ and agent $n_t$, and fix L247–250 to make this decomposition explicit.
>
> > **Can low rank (probabilistic) matrix factorization help?**
>
> Matrix factorization (MF) models utility as $F = MN^\top$, a bilinear function of latent (unknown) item and agent embeddings. Our utility function $f(m_{t,n})$ instead treats item-agent features as observable information, learning a shared unknown function $f$ rather than per-item embeddings. Though MF and our framework aim to approximate the utility function, the key distinction lies in the estimation approach. MF algorithms estimate per-item and per-agent latent vectors, requiring sufficient observations for each item. In our few-copies setting, items appear only $c \ll |\mathcal{M}|$ times, making embedding estimation infeasible. Our approach uses *observable* features of items and agents, generalizing across items without per-item sample accumulation. Our approach is furthermore strictly more general in two ways:
> Non-linear utilities. We extend to non-linear utility functions (Neural-UCB, GP-TS, etc.) well beyond the bilinear MF structure. A bilinear utility is itself a special case of our non-linear framework.
> Fairness-efficiency guarantees. Our algorithms explicitly enforce the fairness-efficiency trade-off via the goodness function $G$ (Eq. 2), with provable fair regret bounds (Theorems 1–3).
>
>
> *We hope that our clarifications will address your concerns. We will incorporate the above responses into the revised version of the paper. If there are no further concerns, we would greatly appreciate your consideration to increase your score.*

---

> > ### Author Rebuttal · Reviewer_1WoJ · 2026-04-05
> >
> > Thanks for your reply, this clears up the confusion. I have increased the score to reflect my assessment.

---

> > > ### Author Response · Authors · 2026-04-08
> > >
> > > We are glad that our responses have fully addressed your concerns, and we greatly appreciate your decision to increase the score to reflect your assessment. Your thoughtful review and suggestions have helped improve the positioning of our paper.
> > >
> > > If you feel that our clarifications further strengthen the work, we would sincerely appreciate your continued support for our paper in the subsequent discussion.

---

### Decision · Program_Chairs · 2026-04-30

**Decision:**

Accept (regular)

**Comment:**

This paper models online fair division with numerous items and few copies per item as a contextual bandit, contributing an OFD-compatible framework and a general reduction that certifies any compatible algorithm with provable fair regret bounds.

The rebuttal was substantive; two reviewers marked concerns fully resolved, and the remaining conceptual question from 74vB was clearly addressed on April 8.

The few-copies setting is novel and well-motivated, and the OFD-compatible reduction is a non-trivial contribution, the history-dependent goodness function breaks standard per-round independence in contextual bandit analysis. The synthetic-only experiments are a legitimate weakness. The authors must follow through on their committed revisions: semi-realistic experiments, prior OFD algorithm baselines, and unified notation throughout.